# Dynamic filopodia are required for chemokine-dependent intracellular polarization during guided cell migration in vivo

Dana Meyen[1], Katsiaryna Tarbashevich[1†], Torsten U Banisch[1,2†], Carolina Wittwer[1], Michal Reichman-Fried[1], Benoît Maugis[1], Cecilia Grimaldi[1], Esther-Maria Messerschmidt[1], Erez Raz[1*]

[1]Institute for Cell Biology, Center for Molecular Biology of Inflammation, Münster University, Münster, Germany; [2]Department of Biological Regulation, Weizmann Institute of Science, Rehovot, Israel

*For correspondence: erez.raz@uni-muenster.de

†These authors contributed equally to this work

Competing interests: The authors declare that no competing interests exist.

**Abstract** Cell migration and polarization is controlled by signals in the environment. Migrating cells typically form filopodia that extend from the cell surface, but the precise function of these structures in cell polarization and guided migration is poorly understood. Using the in vivo model of zebrafish primordial germ cells for studying chemokine-directed single cell migration, we show that filopodia distribution and their dynamics are dictated by the gradient of the chemokine Cxcl12a. By specifically interfering with filopodia formation, we demonstrate for the first time that these protrusions play an important role in cell polarization by Cxcl12a, as manifested by elevation of intracellular pH and Rac1 activity at the cell front. The establishment of this polarity is at the basis of effective cell migration towards the target. Together, we show that filopodia allow the interpretation of the chemotactic gradient in vivo by directing single-cell polarization in response to the guidance cue.

## Introduction

Cell migration is essential for tissue and organ development, for tissue homeostasis and function. The process of migration is also at the basis of pathological conditions such as inflammation and cancer metastasis (*Friedl and Gilmour, 2009*). Migrating cells generate finger-like cellular extensions that contain parallel actin-bundles (*Mallavarapu and Mitchison, 1999*; *Yang et al., 2009*), termed filopodia, that constitute a characteristic feature of metastasizing cells (*Machesky, 2008*; *Arjonen et al., 2011*; *Pan et al., 2011*). It has been proposed that filopodia transport signalling molecules to neighbouring cells (*Sanders et al., 2013*; *Roy et al., 2014*), promote adhesion and are therefore important for generation of traction forces (*Albuschies and Vogel, 2013*; *Fierro-González et al., 2013*) and serve as a sensing organelle (reviewed in [*Wood and Martin, 2002*]). The latter notion arose from functional studies that were carried out in vitro and involved manipulations that could have affected different processes in addition to filopodia formation in chemotrophic growth cones (*Zheng et al., 1996*; *Rajnicek et al., 2006*). Other studies demonstrated an EGF retrograde transport on protrusions of epidermoid carcinoma cells in culture (*Lidke et al., 2005*), although the functional significance of the findings was not explored. Consistent with a role in sensing, filopodia formation was observed in response to VEGF that directs angiogenic sprouting (*Gerhardt et al., 2003*), to FGF that instructs branching morphogenesis in the tracheal system (*Ribeiro et al., 2002*), as well as in response to morphogen signals in *Drosophila* (*Roy et al., 2011*). In the context of group cell

**eLife digest** Some of the cells in an animal embryo have to migrate long distances to reach their final positions; that is to say, to reach the locations where they will participate in the formation of tissues and organs. The migration of cells is also important throughout the entire lifespan of an animal. White blood cells, for example, must be able to move within tissues to search for and fight infections as well as to detect and remove abnormal cells.

The front end of a migrating cell typically protrudes. The back of the cell is then pulled and detaches, which allows the whole cell to move forward. Migrating cells generate thin finger-like projections known as filopodia that have been suggested to help the cell sense their external environments and follow chemical cues. It is not clear what happens to a migrating cell in a living organism if the formation of its filopodia is impaired, or even how filipodia help the normal migration of cells in animals.

To define how filopodia help to guide migrating cells in an animal, Meyen et al. analyzed the migration of cells called 'primordial germ cells' (or PGCs) in zebrafish. These cells form very early on in development of a zebrafish embryo at a position that is far away from their final location (in the testes or ovaries where they will go on to form sperm or egg cells respectively). Meyen et al. revealed that cells that are exposed to the guidance cue (a protein called a chemokine) form more filopodia at their front compared to their rear. The filopodia formed at the cell front also extend and retract more frequently.

Meyen et al. further observed that the specific chemokine that guides the cells can bind to the filopodia and enter the cell. This leads to a signal inside the cell that tells the cell to move in the direction where more of the chemokine is found. Indeed, altering the distribution and number of filopodia around the cell's edge decreases the ability of the primordial germ cells to reach their targets. Together, this work shows that the filopodia at the front end of cells are required for sensing the chemokines that guide cell movement.

Further work is required to understand the mechanism that determines the distribution of filopodia on the surface of migrating cells, and the role of chemokines in the process. Moreover, this work may also be relevant for understanding the migration of cancer cells, because several types of cancer can invade new tissues by following directional cues including chemokines.

migration, inhibiting filopodia formation decreased the migration velocity, yet the cellular basis for this effect has not been further investigated (*Phng et al., 2013*). Similarly, it was suggested that the migration of neural crest cells as streams require filopodia function, since a neuronal crest subset failed to migrate properly in zebrafish mutants that lacked the *fascin1a* gene whose actin bundling function is required for filopodia formation (*Boer et al., 2015*). Nevertheless, the precise consequence of impaired filopodia formation in migrating single cells in vivo and the mechanism underlying their action during normal migration in the context of the intact tissue have thus far not been reported. As a useful in vivo model for exploring the regulation and function of filopodia in cell migration, we employed zebrafish Primordial germ cells (PGCs). These cells perform long-range migration as single cells within a complex environment from the position where they are specified towards their target (*Richardson and Lehmann, 2010*; *Tarbashevich and Raz, 2010*). PGC migration is guided by the chemokine Cxcl12a that binds Cxcr4b, which is expressed on the surface of these cells (*Doitsidou et al., 2002*; *Knaut et al., 2003*). This specific receptor-ligand pair has been shown to control among other processes, stem-cell homing (*Chute, 2006*), cancer metastasis (*Zlotnik, 2008*) and inflammation (*Werner et al., 2013*). Interestingly, similar to other migrating cells types in normal and disease contexts, zebrafish PGCs form filopodia, protrusions whose precise function in guided migration has thus far not been characterized.

We show here that in response to Cxcl12a gradients in the environment, filopodia exhibit polar distribution around the cell perimeter and alter their structural and dynamic characteristics. We demonstrate that PGCs guided by Cxcl12a form more filopodia at the cell front, filopodia that exhibit higher dynamics and play a critical role in receiving and transmitting the polarized signal. Specifically, we show that the short-lived actin-rich filopodia formed at the front of cells migrating within a Cxcl12a gradient are essential for conferring polar pH distribution and Rac1 activity in

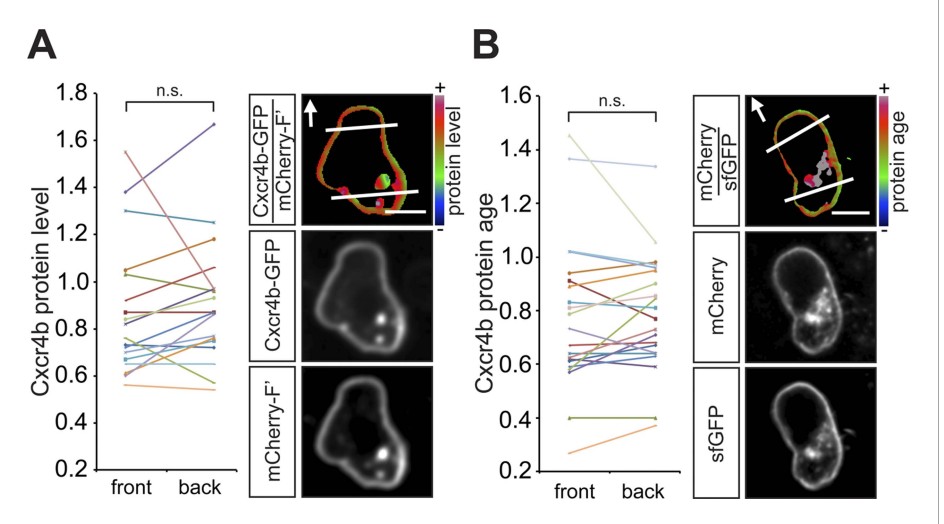

**Figure 1**. In wild type embryos the Cxcr4b receptor is uniformly distributed on the migrating PGC membrane, and its turnover is uniform around the cell circumference. (**A**) A graph showing the Cxcr4b-GFP protein level measured at the front and the back (normalized to the mCherry-F') of individual migrating PGCs under conditions of endogenous Cxcl12a distribution in the embryo (**A**, n = 18). A representative cell is shown, with the areas defined as front and back indicated by lines and the arrows designate the migration direction. Scale bars signify 10 μm. (**B**) A graph showing the protein age (lifetime ratio, see *Figure 1—figure supplement 1*) measured at the front and the back of individual migrating PGCs under conditions of endogenous Cxcl12a distribution in the embryo (n = 20). A representative cell is shown, with the areas defined as front and back indicated by lines and the arrows designate the migration direction. Scale bars signify 10 μm.

The following figure supplement is available for figure 1:

**Figure supplement 1**. Functionality of the Cxcr4b tandem fluorescent timer (tft) in the context of PGC migration.

response to the guidance cue, thus facilitating effective cell polarization and advance in the correct direction. Together, these results provide novel insights into the role of filopodia in chemokine-guided single cell migration, underlining their function in orienting cell migration.

## Results

### The chemokine receptor Cxcr4b is uniformly distributed around the surface of PGCs

Guided towards their target by the chemokine Cxcl12a, zebrafish PGCs generate blebs primarily at the cell aspect facing the migration direction (*Reichman-Fried et al., 2004*). To define the mechanisms that could contribute to the apparent polarity of migrating PGCs, we first measured the distribution of Cxcr4b on the cell membrane around the cell perimeter. Similar to findings in *Dictyostelium discoideum* cells, in which the guidance receptor is evenly distributed around the cell membrane (*Ueda et al., 2001*) and consistent with our previous results (*Minina et al., 2007*), the level of a GFP-tagged Cxcr4b (expressed at low amounts that do not affect the migration) measured at the cell front and its back was similar (*Figure 1A*). Furthermore, the receptor turnover on the plasma membrane, as visualized by a Cxcr4b tandem fluorescent timer (tft) (*Khmelinskii et al., 2012*) expressed in PGCs (*Figure 1—figure supplement 1A–E*), which are directed by the endogenous Cxcl12a gradient (*Figure 1B*), did not reveal a significant difference between the front and the back of the cell. Together, employing the tools described above, we could not detect an asymmetric receptor distribution or differential turnover around the cell perimeter of PGCs in the wild type situation. These findings prompted us to explore qualitative and quantitative differences between the cell front and back, specifically by examining cellular structures that could contribute to the polarity of Cxcr4b signalling.

## Migrating PGCs extend specialized filopodia in the direction of migration

As zebrafish PGCs migrate, they extend protrusions on their circumference that appear like filopodia (*Ahmed et al., 2010*). To determine the role these protrusions play in the process of bleb-dependent single cell migration in vivo, we first studied their characteristics in the course of PGC migration within the three dimensional (3D) cellular environment of the embryo.

Since filopodia are defined as F-actin containing cellular extensions (*Mallavarapu and Mitchison, 1999*), we followed the distribution of F-actin using an Lifeact-EGFP fusion protein. While F-actin was detected in all of the filopodia formed at the cell front, about 50% of those localized to the back of the cell showed no detectable actin signal (*Figure 2—figure supplement 1A*), pointing at differences in actin content between the two populations of filopodia. In the filopodia formed by PGCs migrating in embryos lacking Cxcl12 (*med^NY054*, [*Valentin et al., 2007*]), we observed a higher proportion of filopodia in which actin is not detected (*Figure 2—figure supplement 1B*). Filopodia-like structures devoid of F-actin were previously described (*Phng et al., 2013*; *Yang et al., 2009*).

To characterize the distribution of filopodia in the migrating cells, we imaged PGCs expressing farnesylated EGFP (EGFP-F'). A characteristic wild type PGC is presented in *Figure 2A* and *Video 1*, showing polarized formation of an array of filopodia pointing in the direction of migration (arrow in *Figure 2A*). For a quantitative measure of this finding, we categorized the position of filopodia around the circumference of the PGCs as front, back or side filopodia as illustrated in *Figure 2—figure supplement 2A–A'''*. We found that at any given time point, migrating PGCs show on average $5.6 \pm 0.5$ filopodia, with a strong bias for filopodia generation at the cell front (wild type in *Figure 2C,D*). Whereas filopodia formed preferentially at the cell front, their distribution was independent of that of the forming blebs. Specifically, we found that blebs can form in parts of the cells where filopodia are not present (*Figure 2—figure supplement 3A,B*, red asterisk in 40", *Video 2*) and conversely, filopodia were maintained close to a forming bleb (*Figure 2—figure supplement 3B*, arrowheads in 20", *Video 2*).

By following filopodia dynamics we determined that the extensions persisted for an average of $33 \pm 2.5$ s (wild type in *Figure 3A*, whole cell), from the time of appearance to complete retraction and that on average, they extended to a maximum length of $3.4 \pm 0.1$ μm (wild type in *Figure 3B*, whole cell). Interestingly, the dynamic parameters (persistence and length) were dramatically different between filopodia at the front and back of the cell. That is, while filopodia in the front persisted for $28 \pm 3.5$ s and were $3.0 \pm 0.1$ μm long, filopodia at the back of the cell persisted for $51 \pm 8.2$ s and were $4.3 \pm 0.4$ μm long (wild type in *Figure 3A,B*), thus revealing that filopodia located at the cell front are significantly more dynamic. Importantly, the enhanced dynamics of filopodia at the cell front was not a result of the protrusions being engulfed by blebs that preferentially form at this aspect of the cell. Only 30% of the filopodia at the cell front were engulfed by forming blebs and those that were not engulfed showed a reduced persistence as compared with filopodia formed at the cell back (*Figure 3—figure supplement 1A*).

Together, the formation of dynamic filopodia on migrating PGCs is biased towards the leading edge of the cell, with filopodia formed at the front and back exhibiting significant differences in number, length, persistence and actin content.

## Filopodia formation and dynamics are regulated by the chemokine Cxcl12

Since PGCs migrate towards regions expressing higher levels of Cxcl12a and filopodia formation is biased in the same direction, we sought to determine whether Cxcl12a in the environment influences the generation of filopodia. To this end, we analysed filopodia distribution and their dynamics in *med^NY05* embryos lacking functional Cxcl12a (*Figure 2B* and *Video 3*). PGCs in *med^NY054* embryos show a dramatic increase in the number of filopodia ($11.4 \pm 0.8$, *Figure 2C*, whole cell) relative to wild type ($5.6 \pm 0.5$, *Figure 2C*, whole cell). PGCs in *med^NY054* embryos also show an even distribution of filopodia around the cell circumference, which is in sharp contrast to the polar distribution observed in the wild type embryos (*Figure 2C,D*). Similar results were obtained upon disruption of the Cxcl12a/Cxcr4b signalling pathway by morpholino-mediated Cxcr4b knockdown (*Figure 2—figure supplement 4A,B*). Strikingly, irrespective of their position around the cell perimeter, filopodia of PGCs in *med^NY054* embryos revealed average persistence ($51 \pm 3.1$ s, *Figure 3A*, whole cell) and maximum lengths ($3.9 \pm 0.1$ μm *Figure 3B*, whole cell) similar to filopodia

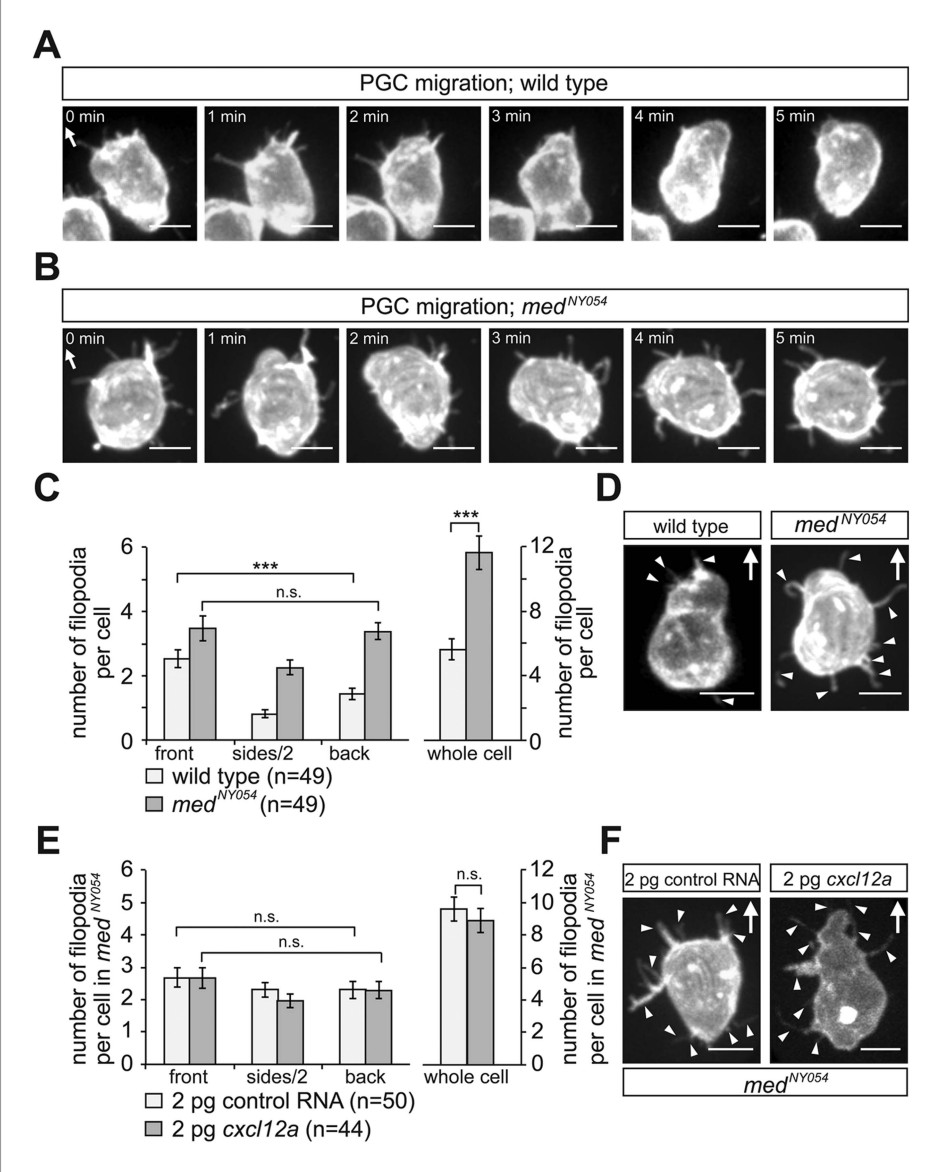

**Figure 2.** The polar positioning and number of filopodia are determined by the Cxcl12a gradient. (**A**) A PGC extending filopodia during migration in wild type embryos (***Video 1***), and (**B**) in medusa (*med^NY054*) mutant embryos lacking Cxcl12a (***Video 3***). (**C**) Filopodia distribution and number in PGCs migrating within wild type and *med^NY054* homozygous embryos (for segmentation see ***Figure 2—figure supplement 2C*** for more details). 'n' indicates the number of cells analysed. (**D**) Examples of a wild type (left) and a *med^NY054* PGC in which the characteristic distribution of the filopodia is indicated by arrowheads. (**E**) Filopodia number and distribution in PGCs migrating within in *med^NY054* homozygous embryos knocked down for Cxcr7b and that express either uniform levels of control RNA (light bars) or Cxcl12a-encoding RNA (dark bars, see also ***Video 4***). (**F**) Examples of PGCs migrating within an environment lacking (left), or containing uniform Cxcl12a (right) in which the characteristic distribution of the filopodia is indicated by arrowheads. In **A**, **B**, **D** and **F** the membrane of the PGCs is labelled with EGFP-F' and arrows indicate direction of movement. Scale bar is 10 μm.

The following figure supplements are available for figure 2:

**Figure supplement 1**. F-actin content in filopodia extended by PGCs.

**Figure supplement 2**. The procedure for cell segmentation, defining the front, back and sides of the cell.

*Figure 2. continued on next page*

*Figure 2. Continued*

**Figure supplement 3**. Filopodia do not appear to play an essential direct role in the generation of blebs.

**Figure supplement 4**. Filopodia distribution and number in PGCs knocked down for Cxcr4b.

located at the back of wild type PGCs (*Figure 3A,B*, wild type). Filopodia of PGCs in *med^NY054* embryos also included a higher frequency of those with no detectable actin, thus exhibiting further similarities to filopodia in the back of wild type cells (*Figure 2—figure supplement 1B*). Next, we examined the effect of a uniform Cxcl12a distribution on filopodia formation. Here, we injected *cxcl12a* RNA into *med^NY054* embryos at a concentration, which is sufficient to interfere with PGC arrival at their target region in wild type embryos, but does not immobilize them (*Figure 3—figure supplement 2A,B* and *Video 4*). In these experiments we co-injected morpholino against *cxcr7b*, the somatic decoy receptor for Cxcl12a (*Boldajipour et al., 2008*), to avoid any local decrease in Cxcl12 levels that would result in gradient formation. Similar to the results observed in *med^NY054* embryos (*Figure 2C*, *med^NYO54* and *Figure 2E*, control), numerous, evenly distributed filopodia appeared on the cell surface of PGCs (*Figure 2E,F*, *cxcl12a*) and their persistence (58 ± 3.0 s), as well as their maximum length (4 ± 0.1 μm) was significantly elevated to the same extent as filopodia in *med^NYO54* embryos (*Figure 3C,D* and *Video 5*). We conclude therefore, that the graded distribution of Cxcl12 determines the number, the distribution and the dynamic behaviour of filopodia around the PGC circumference.

## PGCs extend filopodia towards regions of elevated Cxcl12a levels

Considering that filopodia formation is influenced by the distribution of Cxcl12a, we further monitored the immediate response of PGCs to an artificial Cxcl12a gradient generated in Cxcl12a mutant embryos (*med^NY054*). We focused on the formation of filopodia as the PGCs switch from random to directed migration upon exposure to the chemokine gradient. A steep Cxcl12a gradient was generated in *med^NYO54* embryos by transplanting mCherry-labelled cells that were treated with c*xcr7b* morpholino and expressed either *cxcl12a* or control RNA (*Figure 4—figure supplement 1A*). PGCs exposed to the control transplant generated randomly distributed filopodia (*Figure 4A,C* and *Figure 4—figure supplement 1B*) and were not attracted to the transplanted cells (see *Video 6* control transplant, with snapshots presented in *Figure 4A*, 2/23 cells migrated towards the transplant). In contrast, measuring the angle of filopodia to a Cxcl12a-expressing transplant (*Figure 4D* and *Figure 4—figure supplement 1B*) we observed that PGCs exhibited gradual polarization of filopodia directed towards the attractant source. The biased filopodia orientation was followed by cell polarization (blue arrow in *Figure 4D*) as manifested by the elongation of the cell along the axis of migration, positioning of the golgi to the back of the cell (*Figure 4—figure supplement 1A*) and migration towards the chemokine source (*Video 6*, Cxcl12a transplant, with snapshots presented in *Figure 4B* of a single cell representing 15/17 cells that showed this response). These experiments show that

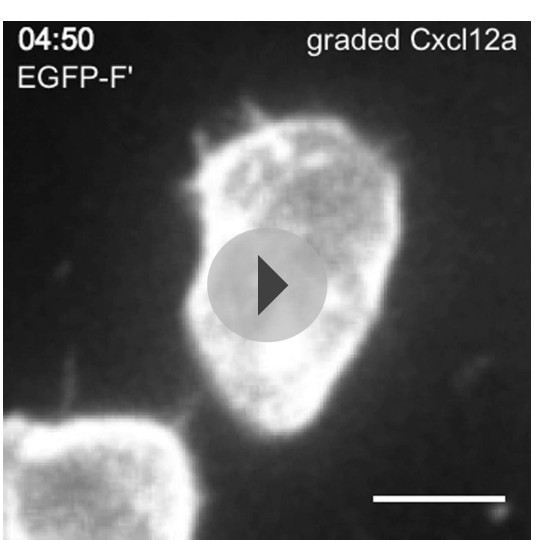

**Video 1.** Migrating PGCs in wild type embryos show enhanced formation of filopodia in the direction of migration. A 10 min time-lapse video of a PGC in *kop-egfp-f'nos3'UTR* embryo was captured using a 63× objective on a Zeiss AxioImager.M2 microscope equipped with a Photometrics camera (Cascade II) and VS-Laser Control. Z-stacks include 29 planes per time point at focal planes 1 μm apart, 10 s interval, 300 ms exposure time with binning one. Time in minutes and seconds. Scale bar represents 10 μm.

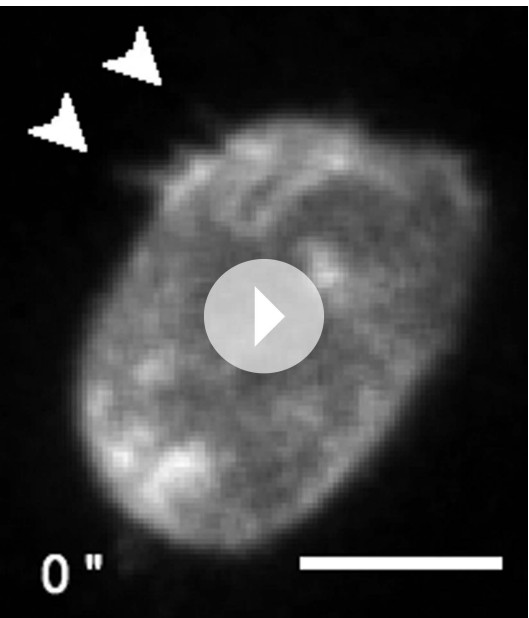

**Video 2.** Filopodia are not directly controlling bleb formation. Bleb-formation (asterisk) can be observed first in close proximity to filopodia (arrowheads, 10–20″) and then in a region devoid of filopodia (40″). A 40-s time-lapse video of a membrane-labelled PGC (in a *kop-egfp-f′nos3′UTR* embryo) using a 63x objective on a Zeiss AxioImager.M2 microscope equipped with a dual view filter (MAG Biosystems), Photometrics camera (Cascade II) and VS-Laser Control. Z-stacks were captured with 29 planes at focal planes per time point, 1 µm apart at 10 s interval, 300 ms exposure time and with binning one. Scale bar represents 10 µm.

PGCs respond to the Cxcl12a gradient first by formation of filopodia oriented towards the Cxcl12a source and then by cell polarization and migration in the direction of the chemokine.

To examine the relevance of these findings for cells migrating within a non-manipulated environment, we monitored the morphology and the behaviour of EGFP-F′ labelled cells in wild type embryos. Interestingly, cells migrating within an unmodified environment behaved similarly to those responding to artificial gradients. Specifically, in cases where migrating PGCs performed sharp turns, polarized filopodia formation preceded the change in migration direction (*Figure 4—figure supplement 2* and *Video 7*). Taken together, the observed PGC response to an artificial and endogenous chemokine gradient is consistent with the idea that filopodia formation constitutes an early response to distribution of Cxcl12 that is followed by cell polarization and directed migration.

## Filopodia interaction with Cxcl12a

Cells respond to graded Cxcl12a in the environment by polarized formation of filopodia (*Figure 2*), which could be important for the orientation of the cell and migration in the direction of the chemokine source. This observation may imply that filopodia could play a role in the transmission of the Cxcl12a signal into the leading edge of the cell by locally increasing the surface area for ligand binding. To address this notion, we overexpressed fluorescently labelled Cxcl12a in cell clones within the embryo (*Figure 5A*) and followed its distribution relative to the cells and particularly, to the filopodia. As shown in *Figure 5B* and *Video 8*, Cxcl12a-Venus foci that were initially detected on the cell surface were subsequently internalized into the cell. Strikingly, a similar interaction between Cxcl12 foci and PGCs was observed on the surface of filopodia (*Figure 5C* and *Video 9*). Binding of Cxcl12 was observed at the tip of a filopodium (red arrowhead in *Figure 5C*) that extended about 12 µm away from the cell body, as well as closer to the cell body (yellow arrowhead in *Figure 5C*). Retraction of the filopodium resulted in dragging of the distant Cxcl12 focus to the cell body and a bleb forming at this location engulfs the closer Cxcl12 focus (*Figure 5C*). The internalization of Cxcl12a-Venus (yellow arrowhead in *Figure 5D*) is consistent with the idea that the ligand interacts with its receptor Cxcr4b on the filopodia membrane and elicits a signal directing cell polarization relevant for cell migration in the direction dictated by the chemotactic gradient. Whereas the filopodia at the cell front are more dynamic and higher in number, consistent with the uniform distribution of Cxcr4b on the cell membrane (*Figure 1A*), we could detect interaction of overexpressed Venus-labelled Cxcl12 with filopodia at the back of the cell as well (*Figure 5—figure supplement 1A–A″*).

## Filopodia at the cell front are required for cell polarization in response to Cxcl12a cues

To determine the functional significance of the Cxcl12a-dependent polarized generation of filopodia, we specifically interfered with their formation and examined the effect on cell polarization. For that purpose, we initially employed the insulin receptor substrate protein 53 (Irsp53), a key protein in the process of filopodia formation (*Ahmed et al., 2010*; *Krugmann et al., 2001*). The Irsp53 protein contains an Inverse-Bin-Amphiphysin-Rvs (I-BAR) domain (*Mattila et al., 2007*) that promotes

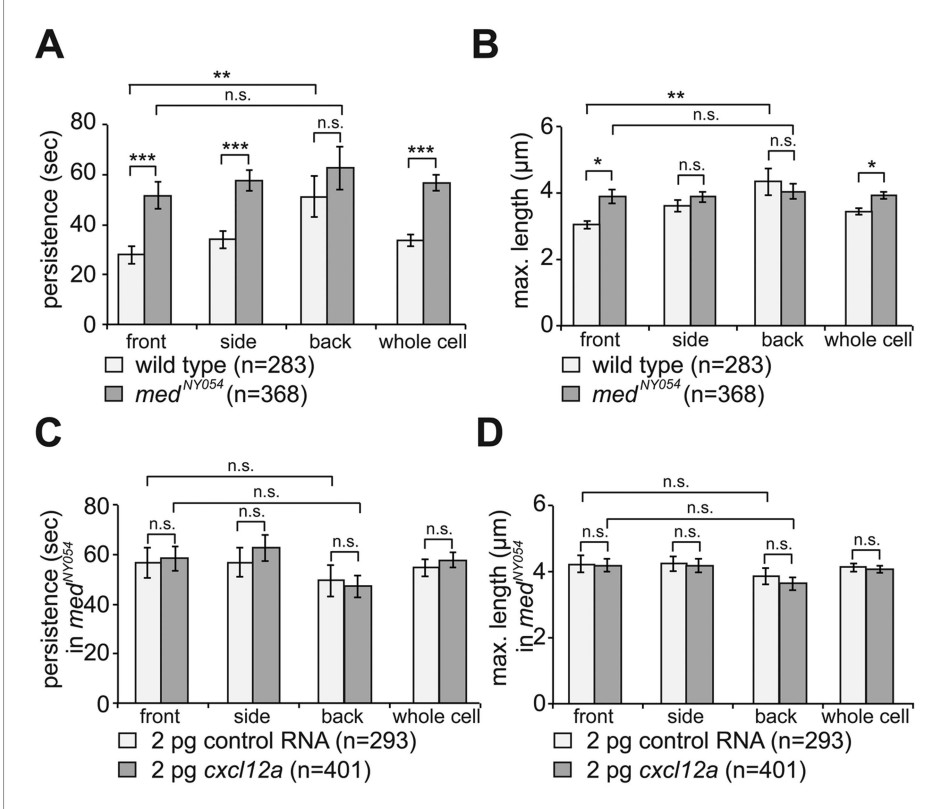

**Figure 3**. The dynamics of filopodia at the cell front are determined by the distribution of Cxcl12a. (**A**) Persistence and (**B**) maximum length of filopodia in PGCs migrating within wild type (light bars) and *med*[NY054] homozygous embryos (dark bars). 'n' indicates the number of filopodia analysed in 10 wild type cells and 8 *med*[NY054] cells over 10 min. (**C**) Persistence and (**D**) maximum length of filopodia in PGCs migrating within *med*[NY054] homozygous embryos knocked down for Cxcr7b and injected with 2 pg control mRNA (light bars) or 2 pg *cxcl12a* mRNA (dark bars). 'n' indicates the number of filopodia in 10 cells over 10 min.

The following figure supplements are available for figure 3:

**Figure supplement 1**. The enhanced dynamics of filopodia at the cell front is independent of bleb formation at this aspect of the cell.

**Figure supplement 2**. Properties of PGCs migrating in embryos expressing a low concentration of uniform Cxcl12a.

membrane deformation and includes actin binding sites (*Yamagishi et al., 2004*). The protein also contains a partial CRIB motif for Cdc42-GTP mediated activation (*Govind et al., 2001*) and an SH3 domain for interaction with additional filopodia regulators such as Mena (*Krugmann et al., 2001*), Eps8 (*Vaggi et al., 2011*), mDia1 (*Goh et al., 2012*), WAVE2 (*Goh et al., 2012*), N-WASP (*Oh et al., 2013*), Lin7 (*Crespi et al., 2012*) and WIRE (*Misra et al., 2010*) (*Figure 6A*). We found that the expression pattern of *irsp53* mRNA in zebrafish embryos spans the relevant early stages during which PGCs migrate. Specifically, *irsp53* RNA is maternally provided and ubiquitously present in 1-hr post fertilization (hpf) embryos, as well as at later stages (6 and 10 hpf, *Figure 6B*). We also found that an Irsp53-mCherry fusion protein (*Figure 6A*) was expressed in the cytoplasm and importantly, in filopodia (*Figure 6C*, upper panel and *Figure 6—figure supplement 1A*), particularly in those located at the cell front. The expression pattern of *irsp53* RNA and the subcellular localization of the Irsp53 fusion protein to filopodia are therefore consistent with the possibility that Irsp53 plays a role in filopodia formation in migrating PGCs.

To assess filopodia function in migrating PGCs, we set out to disrupt their formation by expressing a dominant-negative (DN) form of the Irsp53 protein. The dominant-negative form of the protein harbours mutations in the four actin-bundling sites (*Millard et al., 2005*; *Lim et al., 2008*) within the

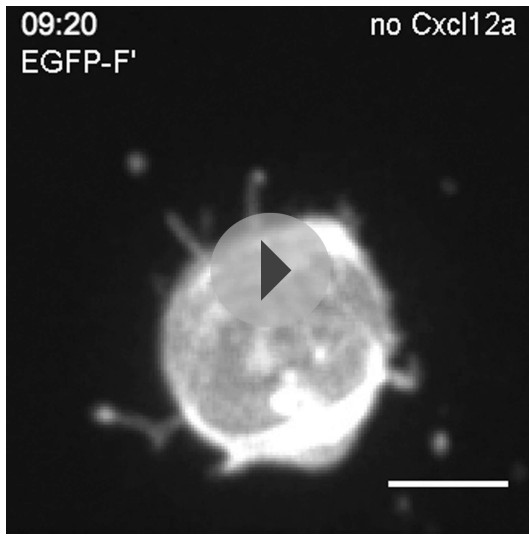

**Video 3.** Migrating PGCs in *medusa* (*med^NY054*) mutant embryos that lack Cxcl12a show enhanced apolar filopodia formation. A 10 min time-lapse video of a PGC in *med^NY054* homozygous embryo was captured using a 63× objective on a Zeiss AxioImager.M2 microscope equipped with a Photometrics camera (Cascade II) and VS-Laser Control. Z-stacks include 29 planes per time point at focal planes 1 µm apart, 10 s interval, 300 ms exposure time with binning one. Time in minutes and seconds. Scale bar represents 10 µm.

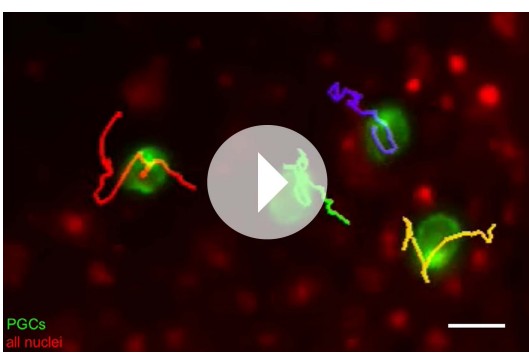

**Video 4.** Low concentration of uniformly expressed Cxcl12a affects cell migration, but does not immobilize the cells. Migration tracks of PGCs in *med^NY054* homozygous embryos knocked down for Cxcr7b, injected with *mcherry_h2b_globin3′UTR* mRNA (mCherry labelling nuclei of all cells) and either with 2 pg control mRNA (first video) or 2 pg *cxcl12a* mRNA (second video). Tracks represent 58 min of development in 7 hpf embryos. Snapshots captured at 2-min intervals with 300 ms exposure time at three focal planes (15 µm apart) using a 10× objective to generate the Z-stacks on a Zeiss AxioImager.M1 microscope equipped with a Photometrics camera (CoolSNAP ES2). Time in minutes and scale bar represents 50 µm.

I-BAR domain (*Figure 6A*). A DN Irsp53-mCherry protein version expressed in PGCs was present primarily in the cell body, rather than in filopodia (*Figure 6C*, lower panel, *Figure 6—figure supplement 1*). Moreover, expression of the DN Irsp53 version in PGCs resulted in a significant decrease in filopodia number at the cell front (*Figure 6D,E*). However, inhibition of Irsp53 function had no effect on filopodia formation in *med^NY054* embryos (*Figure 6F,G*), in which the filopodia exhibit posterior characteristics (*Figure 3A*). The inhibition of Irsp53 function specifically affected filopodia formation rather than influencing cell migration in general, as this treatment had no effect on the formation of blebs at the PGC front in embryos lacking the guidance cue (*Figure 6—figure supplement 2A*). These results provide evidence that Irsp53 is required for filopodia formation in response to Cxcl12a gradient and can serve as a specific reagent for interfering with their formation.

In a second line of experiments attempting to affect filopodia formation we employed the actin filament associated protein 1L1 (AFAP1L1). AFAP1L1 was shown to play a role in tumour metastasis and to constitute a cancer prognostic marker (*Furu et al., 2011*), to be localised to invadosomes and along with cortactin, to podosomes (*Snyder et al., 2011*). AFAP1L1 contains two PH domains that serve as a link to the membrane, a serine-threonin-rich substrate domain (SD) in between, a leucine zipper (Lzip), an actin-binding domain (ABD) at the C-terminus, as well as one SH3 and two SH2 domains (*Figure 7A*) (*Snyder et al., 2011*). While AFAP1L1 had not been previously implicated in filopodia formation, its subcellular localization in PGCs was intriguing and prompted our following investigation. Namely, when an mCherry-tagged version of Afap1L1a (*Figure 7A*) was expressed in PGCs, it was found to be specifically enriched at the base of most filopodia (84 ± 9.7%) at the level of the cell cortex (*Figure 7C*), suggesting a function for the protein in filopodia formation or maintenance. We subsequently ascertained that *afap1l1a* RNA is expressed at the time at which PGCs migrate towards their target and showed that *afap1l1a* RNA is maternally provided and is uniformly expressed during early embryonic development (*Figure 7B*).

To test whether Afap1L1a could serve as a tool for altering filopodia formation, we overexpressed the protein in PGCs. Remarkably, overexpression of the protein resulted in a significant increase in filopodia number as compared to control cells (*Figure 7D*, whole cell). Furthermore, this

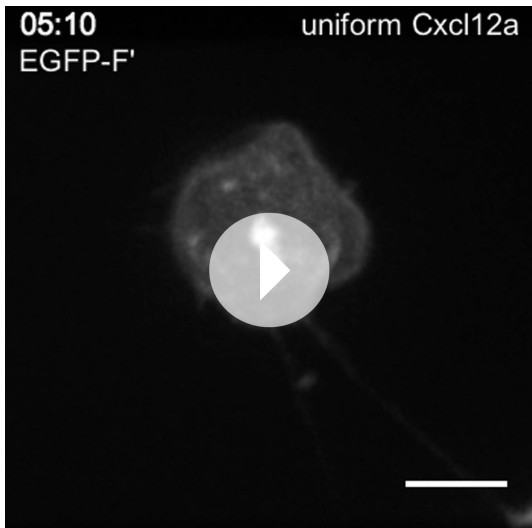

**Video 5.** Uniform Cxcl12a in the environment induces apolar formation of numerous long and persisting filopodia on the PGC surface. A 10 min time-lapse video of a PGC in *med*[NY054] homozygous embryo knocked down for Cxcr7b and that expresses uniform levels of Cxcl12a-encoding RNA was captured using a 63× objective on a Zeiss AxioImager.M2 microscope equipped with a Photometrics camera (Cascade II) and VS-Laser Control. Z-stacks include 29 planes per time point at focal planes 1 μm apart, 10 s interval, 300 ms exposure time with binning one. Time in minutes and seconds. Scale bar represents 10 μm.

treatment changed the distribution of the filopodia significantly, resulting in a decrease in filopodia number at the cell front and an increase in number at the sides and back of the cells (*Figure 7D,E*), effectively reversing the normal distribution of filopodia in the cell. Based on their increased persistence and length (*Figure 7—figure supplement 1A,B*), the filopodia formed by cells overexpressing Afap1L1a were of the type generated at the back of PGCs. Similar to the Irsp53 effect, overexpression of Afap1L1a did not significantly alter bleb formation at the PGC front in embryos lacking the guidance cue (*Figure 7—figure supplement 2A*). Taken together, these findings suggest that Afap1L1 could serve as a specific tool for manipulating the formation of filopodia in PGCs.

Since manipulating Irsp53 and Afap1L1a function in PGCs altered filopodia distribution and number, we were in a position to probe into the functional significance of the polar formation of filopodia as potential sensors of the guidance cue. To this end, we made use of our recent findings showing that in response to a graded Cxcl12a concentration in the environment, zebrafish PGCs exhibit increased pH at the cell front (7% higher on the logarithmic pH scale, reflecting difference of 15% at the level of $H^+$ concentration). This polar pH elevation could control the activity of various proteins important for cell motility (*Stock and Schwab, 2009*) and in the case of PGCs, was shown to be essential for a local increase in the activity of the small GTPase Rac1 and thus, for directing actin polymerization to the cell front (*Tarbashevich et al., 2015*). The loss of the Cxcl12a gradient leads the loss of the difference in pH between the cell front and its rear (*Tarbashevich et al., 2015*). We reasoned that if filopodia were indeed important for Cxcl12 sensing, they should be functionally linked to pH elevation and Rac1 activity at the front of PGCs. To test this possibility, we first performed FRET-based pH measurements in cells in which filopodia formation was manipulated as described above. Strikingly, the reduction of filopodia formation at the cell front reduced (in the case of Irsp53), or abolished (in the case of the Afap1L1a) the increased pH at the leading edge (*Figure 8A*). Conversely, inhibiting the relative elevation of pH at the cell front by inhibiting the translation of the Carbonic anhydrase 15b (*Tarbashevich et al., 2015*) had no effect on filopodia formation (*Figure 8—figure supplement 1A,B*). To determine whether the reduction of filopodia formation at the cell front had any effect on Rac1 activity, we expressed a FRET-based Rac1 activity reporter in cells overexpressing Afap1L1a. Importantly, as compared with control cells, Rac1 activity was significantly reduced in these cells (*Figure 8B*). A milder effect on the difference between the front and rear pH, as induced by DN Irsp53 (*Figure 8A*) could affect different proteins important for cell motility (*Stock and Schwab, 2009*), but was not associated with a significant reduction in Rac1 activity as judged by the FRET level. Consistent with the idea that filopodia function upstream of the Cxcl12a signalling cascade, expression of a constitutively activated version of Rac1 (Rac1[V12]) in germ cells (that stop migrating as a result) had no effect on filopodia formation beyond the increase in their numbers observed in cells rendered immotile (*Figure 8—figure supplement 1C,D*). These results suggest that when filopodia function at the cell front is hindered, the ability of the PGCs to effectively sense the chemokine gradient is impaired, as reflected by the lack of elevated pH and by the reduced Rac1 activity at the leading edge. These results are also consistent with the idea that as a cellular structure, filopodia function in controlling intracellular pH at the cell front and are formed independent of it.

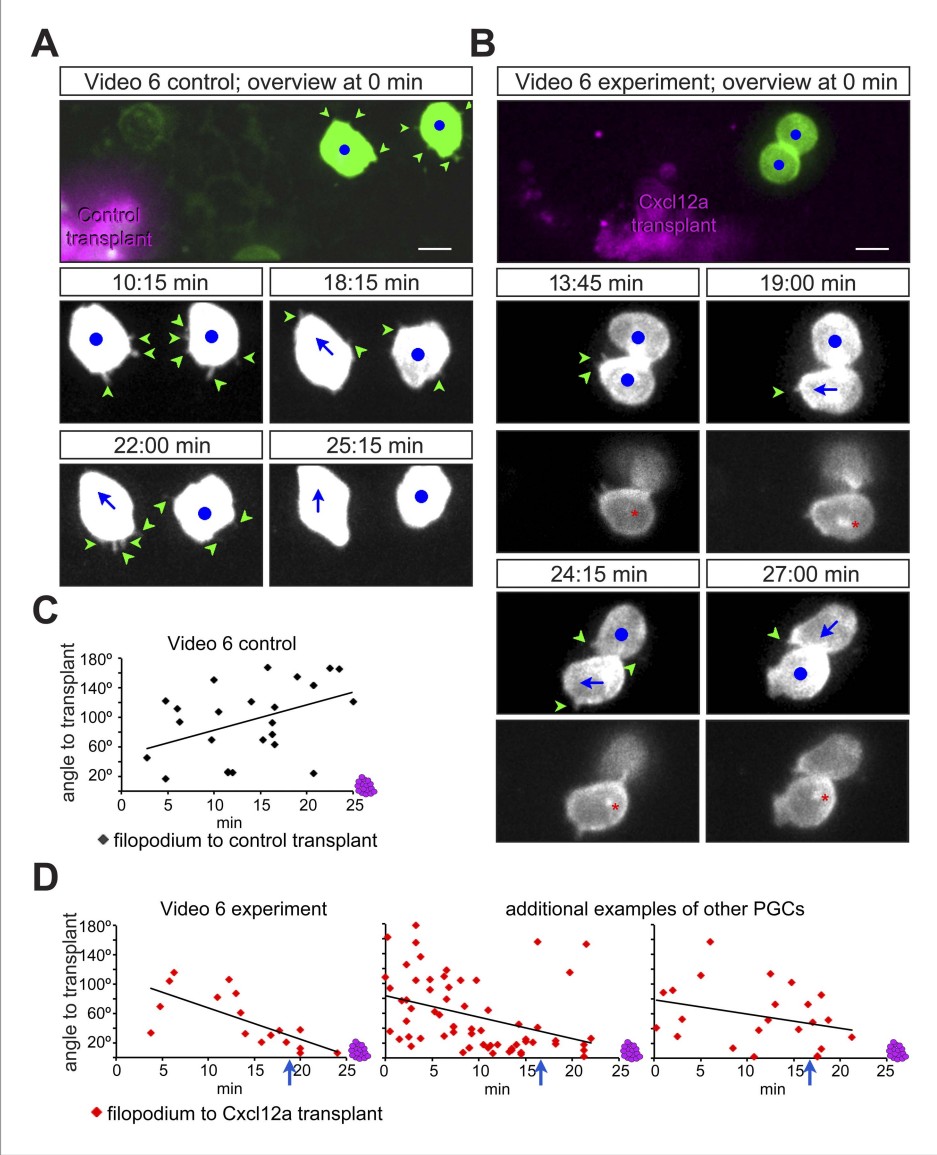

Figure 4. PGCs extend filopodia towards the chemokine source prior to cell polarization and directed migration towards the attractant. (**A**, **B**) The cellular behaviour of PGCs (green) in response to transplanted control cells (magenta in **A**) or to Cxcl12a-expressing cells (magenta in **B**). Upper panels show the cells immediately after transplantation and lower panels show snapshots from *Video 6* presenting the behaviour of the cells in the following 28 min. In **B**, additional images present the polar position of the golgi (red asterisk, labelled by EGFP-F', as defined in *Figure 4—figure supplement 1A*) at the back of the cell. Green arrowheads mark filopodia, blue dots indicate no migration and blue arrows show the direction of PGC movement. Scale bar is 10 µm. (**C**, **D**) The angle of filopodia orientation relative to the position of transplanted cells (located at 0°, see also *Figure 4—figure supplement 1B*) in the case of a control transplant (**C**) and with respect to Cxcl12a expressing transplant (**D**, three examples) over 25 min. Blue arrows signify the time of morphological cell polarization and movement.

The following figure supplements are available for figure 4:

**Figure supplement 1**. Polarization of PGCs encountering an artificially generated Cxcl12a gradient.

**Figure supplement 2**. PGCs extend filopodia in the direction of migration prior to polarization and actual onset of migration.

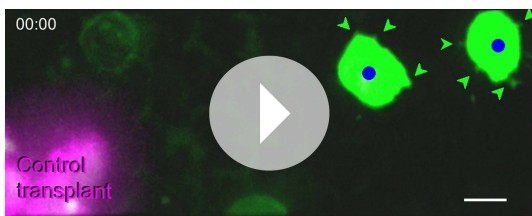

**Video 6.** PGCs extend filopodia towards the chemokine source, prior to their morphological polarization and directed migration towards cells producing the attractant. Cells from 4 hpf *med*^NY054 homozygous embryos, in which Cxcr7b expression was inhibited and which express mCherry-F' (red) and either control RNA (part control transplant) or express *cxcl12a* RNA (part Cxcl12a transplant) were transplanted into 6 hpf host *med*^NY054 homozygous embryos. The PGCs in the host embryos were labelled with EGFP-F' and their reaction to the transplant was monitored on a Zeiss AxioImager.M1 microscope equipped with a dual view filter (MAG Biosystems), Photometrics camera (Cascade II) and VS-Laser Control. Acquisition of the 28 min time-lapse video started immediately following the transplantation using a 40× objective. Z-stacks were captured with 24 planes per time point, at focal planes 4 µm apart, 15 s interval, 300 ms exposure time with binning one. Green arrowheads mark the position of filopodia, red asterisks mark the position of the golgi, blue dots indicate no migration and blue arrows indicate morphological PGC polarization and movement. Time in minutes and scale bar represents 10 µm.

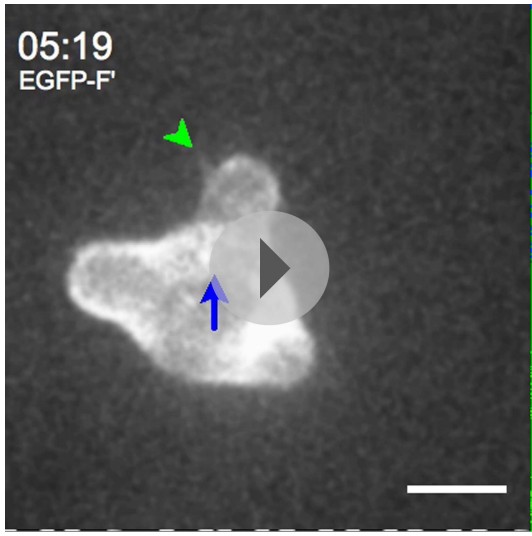

**Video 7.** PGCs extend filopodia in the direction of migration, prior to polarization and actual onset of migration. 10 min time-lapse videos of PGCs in wild type embryos were captured using a 63× objective on a Zeiss AxioImager.M2 microscope equipped with a Photometrics camera (Cascade II) and VS-Laser Control. Z-stacks include 29 planes per time point at focal planes 1 µm apart, 11 s interval, 300 ms exposure time with binning one. Time in minutes and seconds. Scale bar represents 10 µm.

Together, these experiments show that the filopodia at the cell front are important for the downstream signalling by Cxcr4b and thus, for cell polarization and establishment of the cell front in response to chemokine cues.

## Filopodia at the cell front are required for chemokine-guided cell migration

To determine whether inhibition of front filopodia formation affects the directed migration of PGCs, we examined manipulated cells for their migration tracks and arrival at their target. Interestingly, PGCs expressing the DN Irsp53 protein exhibited a significant reduction in displacement (the shortest distance between the start and end points) as a result of reduced track straightness (the degree of changes in path direction) (*Figure 9A,B*). The inhibition of front filopodia mediated by DN Irsp53 appears therefore to impair sensing of the Cxcl12 guidance cue as reflected by the altered characteristics of the migration route. The specific role of front filopodia in the response to the chemokine gradient is further highlighted by the finding, that inhibition of Irsp53 function in *med*^NY054 embryos lacking functional chemokine had no effect on the migration tracks of PGCs (*Figure 9—figure supplement 1A,B*). These results also emphasize the fact that Irsp53, which specifically affects filopodia formation at the cell front, is not essential for the migratory behaviour per se, but is rather important in the context of chemotaxis.

Alterations in the migration track properties were observed yet to a greater extent, upon overexpression of the Afap1L1a in PGCs. Specifically, overexpression of the Afap1L1a that led to a reversal of filopodia distribution with predominance of filopodia in the back, also led to a reduction in cell displacement, track straightness and speed (*Figure 9C,D*).

We conclude therefore, that tight regulation of filopodia formation and distribution around the cell surface is essential for effective guided PGC migration. The defects in the dynamic characteristics of cell migration manifested in the cell tracks of manipulated PGCs resulted in an increased proportion of chemokine-guided cells that failed to reach their target at 24 hpf (*Figure 9E–H*).

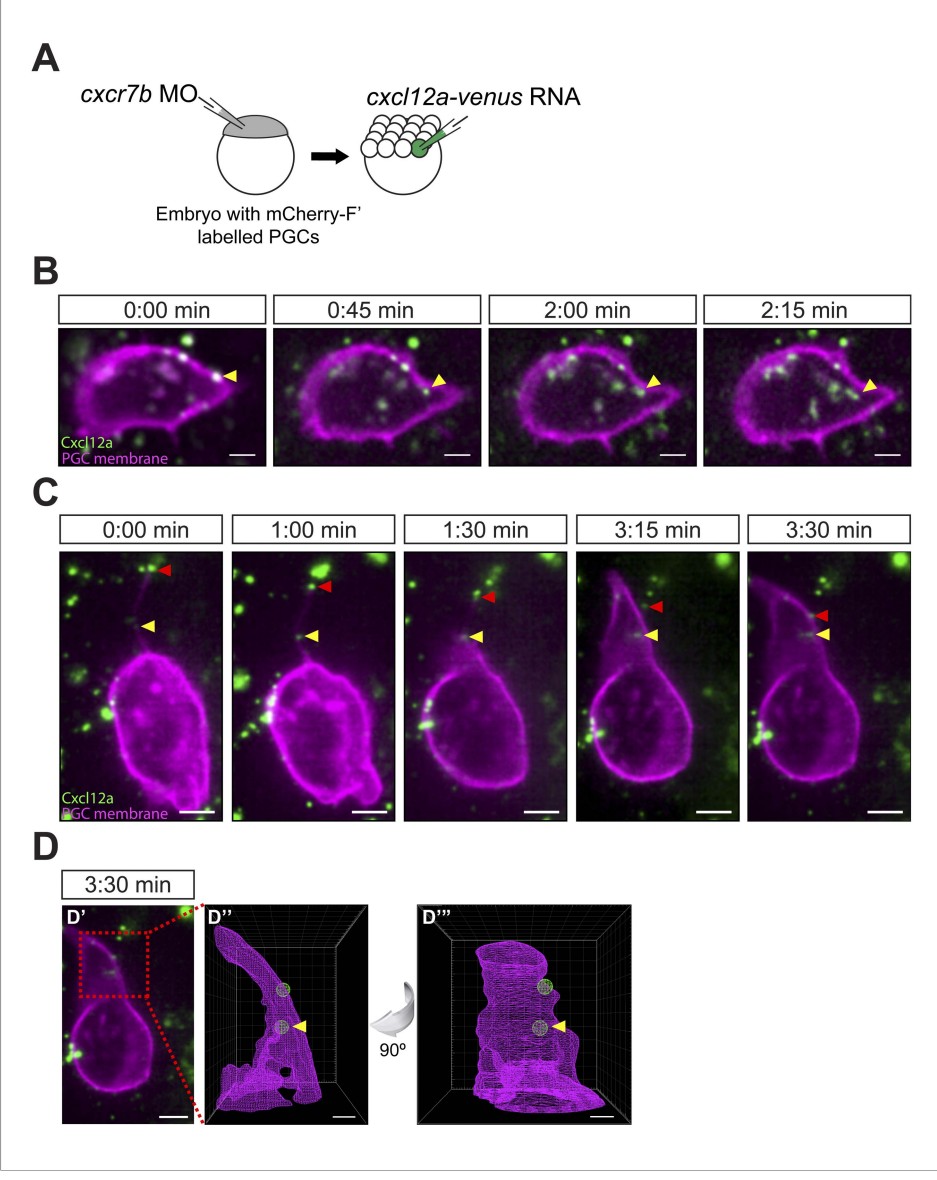

**Figure 5**. Cxcl12a internalization and interaction with filopodia. (**A**) Schematic experimental setup. Cxcr7b function was knocked down in embryos, in which PGCs express mCherry on their membrane. At 16-cell stage, these embryos were injected with *cxcl12a-venus* RNA directed into a corner cell for a mosaic expression of the chemokine. (**B**) Cxcl12a (green) is bound to the PGC membrane (magenta) and internalizes into the cell. Snapshots from *Video 8*, showing an optical section of a PGC (a Z-projection of two 1-μm-slices). An arrowhead points at an internalizing Cxcl12a spot. Scale bar is 5 μm. (**C**) Snapshots from *Video 9*, showing an optical section of a PGC (a Z-projection of 4 1-μm-slices) with Cxcl12a (green) interaction seen on the filopodium (magenta). The red arrowhead indicates a Cxcl12a spot bound to the tip of the retracting filopodium and the yellow arrowhead points at Cxcl12a, which is bound to the filopodium closer to the cell body and is then engulfed by the cell (1:30–3:30 min). Scale bar is 5 μm. (**D**) PGC from panel **C** at 3:30 min (**D'**) where the area magnified in **D"** and **D'''** is delineated in a red box. (**D"** and **D'''**) A 3D wire presentation of the magnified cell surface area (magenta) and the relevant Cxcl12 foci (green). (**D"**) A 3D image orientated as the original panel in **C** and (**D'''**) is horizontally rotated (90° clockwise) to visualize internalization of Cxcl12a. Scale bar is 2 μm.

The following figure supplement is available for figure 5:

**Figure supplement 1**. Cxcl12a interaction with filopodia at the front and back of the cell.

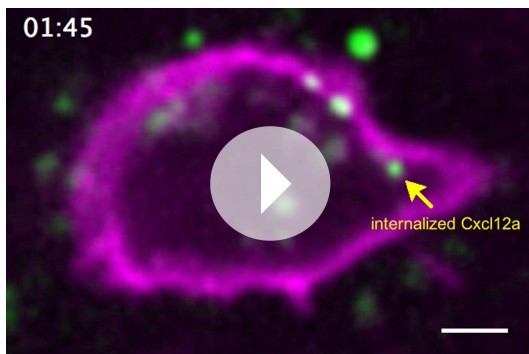

**Video 8.** Cxcl12a internalization. Cxcl12a (green) is bound to the PGC membrane (magenta) and internalizes into the cell. The video was captured using a 63× objective on a Zeiss AxioImager.M2 microscope equipped with a Photometrics camera (Cascade II) and VS-Laser Control at 15 s intervals for 2:15 min and shows an optical section of a PGC (a Z-projection of two 1-μm-slices). Arrows indicate a bound and an internalizing Cxcl12a spot. Scale bar is 5 μm.

## Discussion

In this work we provide evidence that filopodia are asymmetrically formed in response to a chemokine gradient and in turn, play a critical role in polarization of cells and their response to directional cues.

We found that filopodia properties differ in the front from the back of the cell, such that front filopodia are larger in number, turn over more rapidly, are shorter and contain more F-actin. These differences in filopodia distribution and dynamics are governed by the graded distribution of the chemokine in the environment. The basis for this notion is the finding that filopodia formation in PGCs migrating within embryos lacking Cxcl12a was altered in a similar manner to that observed under conditions where the chemokine is uniformly expressed.

The precise function of the front filopodia emerges when PGCs are experimentally exposed to a focused Cxcl12a source: there, filopodia formation appears to constitute an early response of the cells to the graded distribution of the chemokine. The formation of filopodia precedes further polarization such as directed bleb formation and onset of migration towards the chemoattractant source. A similar phenomenon was observed in neuronal growth cones, which respond to a glutamate gradient by an asymmetric distribution of filopodia prior to turning, in an in vitro setting (*Zheng et al., 1996*). Similarly, endothelial tip cells of the early postnatal retina in mice respond to a source of VEGF by filopodia formation (*Gerhardt et al., 2003*) and RhoD-activated fibroblasts extend cytoneme-like protrusions towards an FGF source in vitro (*Koizumi et al., 2012*). Whereas these observations were consistent with the idea that polarized filopodia formation constitutes a response to a guidance cue, the relevance for directed cell migration was not known. Our finding that the chemokine appears to bind and be transported on the filopodium, followed by internalization, is consistent with the idea that filopodia function in Cxcl12a uptake and in increasing the surface relevant for this action at the cell front. In this study, we carried out specific manipulations affecting filopodia formation and show an adverse effect on the ability of migrating single cells to respond to Cxcl12a by directional migration. The effect of specifically decreasing filopodia formation at the cell front (by inhibiting Irsp53 function) resulted in inefficient propagation of the external signal into the cell as evidenced by the strong reduction in the difference between the pH at the cell front and its back. In the same direction, overexpression of the Afap1L1 protein erased the normal front-back polarity with respect to filopodia distribution. In the absence of polar distribution of filopodia in the direction of the attractive cue, the cells could not effectively

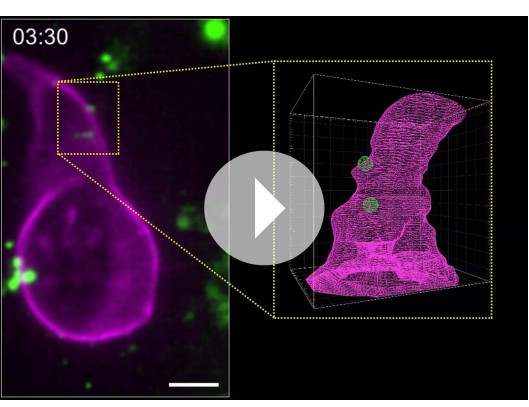

**Video 9.** Cxcl12a interaction with filopodia. Cxcl12a (green) is bound to a filopodium and internalizes into the cell. An optical section of a PGC (a Z-projection of 4 1-μm-slices) is shown with Cxcl12a (green) interaction observed on the filopodium (magenta). The video was captured with a Lightsheet Z1 microscope (Zeiss, Germany) at 15 s intervals for 3:30 min The red arrowhead indicates a Cxcl12a spot bound to the tip of the retracting filopodium and the yellow arrowhead points at Cxcl12a, which is bound to the filopodium more proximally and is then engulfed by the cell. Wire presentation visualizes Cxcl12a inside the cell. Scale bar is 5 μm.

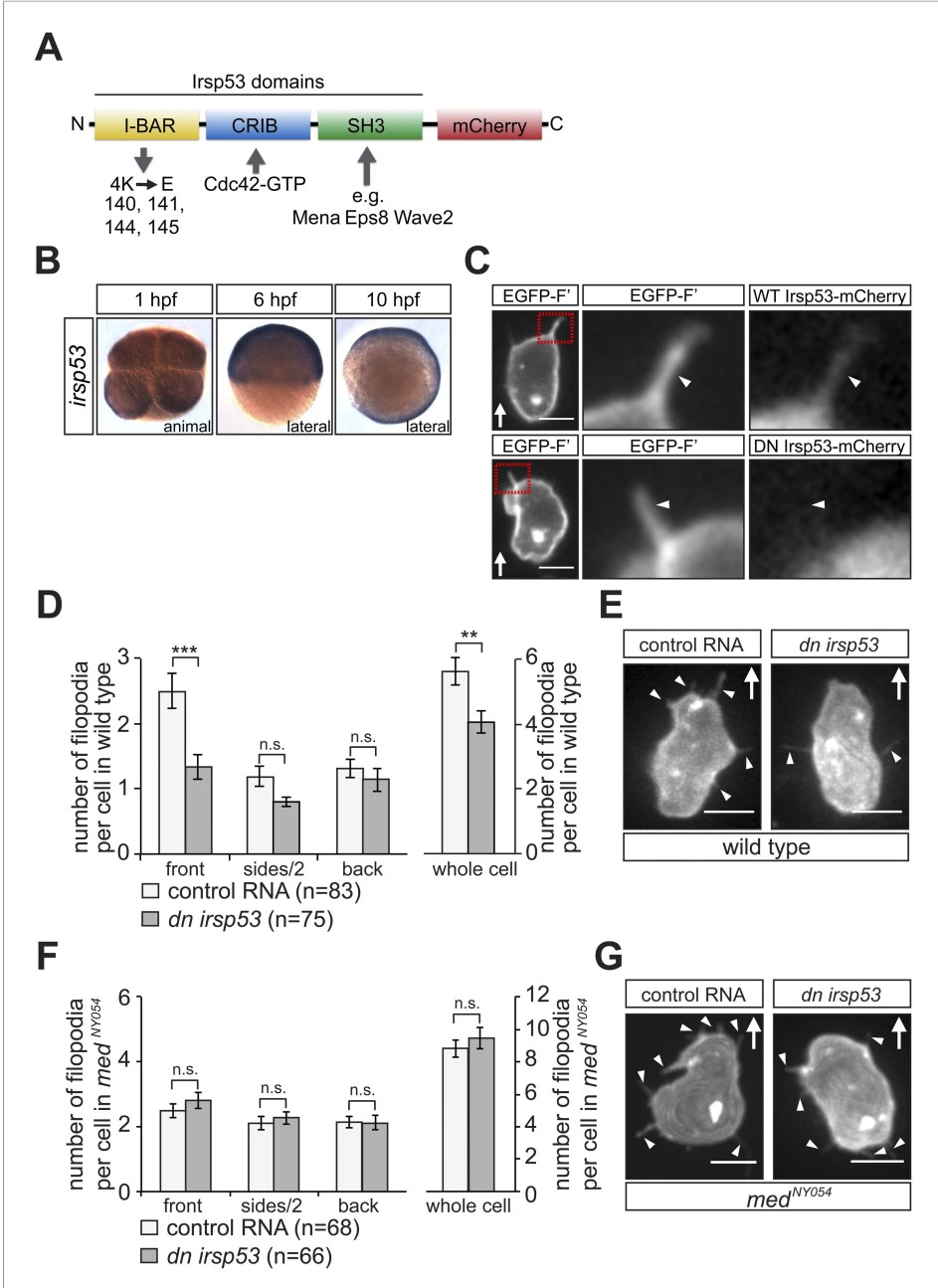

**Figure 6**. *irsp53* RNA expression, Irsp53 protein localization and the role of the protein in filopodia formation. (**A**) Schematic structure of the Irsp53 protein domains. The position of the mCherry fluorophore fusion at the C-terminus of the protein is presented, proteins interacting with the SH3 and CRIB domains are indicated and the mutations introduced into the I-BAR domain to generate the dominant-negative (DN) Irsp53 version are marked. (**B**) Ubiquitous expression of the *irsp53* RNA in 1, 6 and 10-hpf embryos. (**C**) A single plane of PGCs expressing EGFP-F' and an Irsp53-mCherry protein fusion showing localization of Irsp53 to filopodia (upper panel), while the dominant-negative Irsp53 protein is not found in the filopodia (lower panel). Rectangles delineate the area of magnification shown in the right panels. (**D**) Reduction of filopodia number at the cell front in PGCs in wild type embryos expressing the DN Irsp53 protein (dark bars) relative to control PGCs (light bars). 'n' indicates the number of cells analysed. (**E**) Examples of a control (left) and a *dn irsp53*-expressing (right) PGCs in wild type embryos. (**F**) Expression of *dn irsp53* in PGCs of *med^NY054* homozygous embryos shows no effect on filopodia number and distribution around the cell perimeter (dark bars) as compared with control PGCs (light bars). (**G**) Examples of a control (left) and a *dn irsp53*-expressing (right) *Figure 6. continued on next page*

*Figure 6. Continued*

PGCs in *med^NY054* homozygous embryos. 'n' indicates the number of cells analysed, arrows show direction of cell migration and arrowheads mark filopodia. Scale bar is 10 μm.

The following figure supplements are available for figure 6:

**Figure supplement 1**. Irsp53 localization within filopodia.

**Figure supplement 2**. Inhibition of Irsp53 does not affect bleb formation at the cell front.

---

detect the graded chemokine cue, establish a stable front and thus, did not elevate the pH and Rac1 activity at the leading edge. These effects on the characteristic front-back cell polarity impaired the directional migration in response to a guidance cue, culminating in ectopic localization of the cells at the end of the migration process.

Taken together, this work presents important insights into the mechanism by which dynamic filopodia located at the front of migrating cells participate in the transmission of the guidance cue to allow directional migration (*Figure 10*). We suggest that following the activation of Cxcr4b by Cxcl12a, the scaffold protein Irsp53 is activated to promote the formation of dynamic filopodia at the cell front. These filopodia extend in the direction of higher chemokine concentration and expand the surface for Cxcl12a binding, thereby further enhancing Cxcr4b signalling at the cell front and effectively enhancing the absolute differences in ligand levels detected along the front-back axis of the cell. We suggest that the positive feedback loop presented in *Figure 10* allows the PGCs to interpret shallow gradients by forming a front that is more sensitive to the guidance cue. The uneven receptor activation along the back–front axis of the cell that includes the filopodia is a prerequisite for the establishment of the intracellular elevated pH at the cell front that facilitates actin polymerization at this location (*Tarbashevich et al., 2015*) leading to functional polarization of the cell. Under conditions of no, or uniform activation of Cxcr4b a cell front is not established. In such cases, the positive feedback loop that normally stabilizes the front in one aspect of the migrating cell is not present, affecting cell polarity and the features of migration in the direction of the chemokine source.

An important question that remains to be elucidated is the mechanism by which Cxcl12/Cxcr4b signalling activates Irsp53 function, leading to formation of the Irsp53-dependent filopodia at the cell front. CDC42 was shown to be active at the leading edge of migrating cells and in neuronal growth cones (*Etienne-Manneville and Hall, 2002*; *Nalbant et al., 2004*), and IRSp53 was shown to be one of the effectors of CDC42 (*Govind et al., 2001*; *Krugmann et al., 2001*). A direct or indirect effect of Cxcl12 binding to Cxcr4 on the activation of Cdc42 and thus of Irsp53 would be an interesting option to explore.

The finding that at least two populations of filopodia formed on the surface of migrating PGCs and that these populations exhibit differences concerning their distribution and dynamic properties are reminiscent of the findings concerning cytonemes in *Drosophila* (*Roy et al., 2011*). In this case, tracheal cells were shown to generate morphologically different populations of cytonemes, each responding specifically to different secreted signals.

The effect the reversed formation of filopodia had on cell migration speed in PGCs manipulated for Afap1L1 function (*Figure 9D*) raises the option that these protrusions contribute to cell motility in normal cells. This possibility is supported by the observations that some filopodia adhere to somatic cells, appear to be under tension and can pull particles towards the cell (*Kress et al., 2007*). Upon laser-mediated cutting of a filopodium of this kind, its distal tip remained attached to a cell in the environment, while the proximal part attached to the cell rapidly retracted (see *Video 10*). It would be interesting to evaluate the significance of this finding relative to the role we demonstrated concerning chemokine signalling and cell polarization.

The function of filopodia in decoding the Cxcl12a gradient in the context of PGC migration must bear on cancer cell invasion, since Cxcl12 is known to play a key role in metastasis of several cancer types (*Vila-Coro et al., 1999*; *Singh et al., 2004*; *Sun et al., 2010*). Furthermore, proteins involved in filopodia formation such as Fascin, Eps8 and Irsp53 have been shown to be associated with increased invasiveness of cancer cells (*Matoskova et al., 1995*; *Funato et al., 2004*; *Machesky and Li, 2010*). The translation of Cxcr4b activation into filopodia formation constitutes the first morphological step in PGC polarization in response to the chemokine. It would thus be interesting to examine this

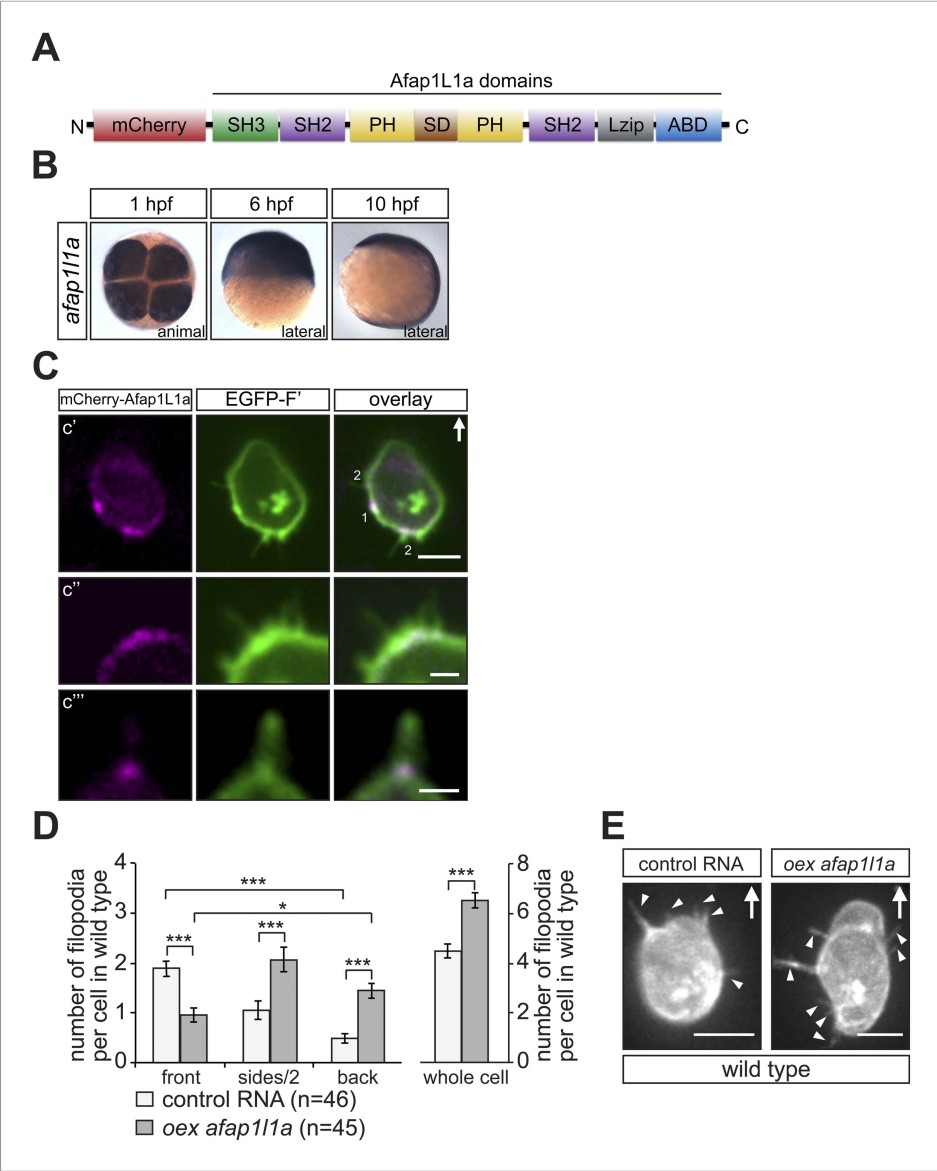

**Figure 7**. *afap1l1a* RNA expression, Afap1L1a protein localization and the role of the protein in filopodia formation.
(**A**) Schematic structure of the Afap1L1a protein domains including a serine-threonine-rich substrate domain (SD) flanked by two PH domains, a leucine zipper (Lzip), an actin-binding domain (ABD), two SH2 and one SH3 domains. mCherry fluorophore was fused to the N-terminus of Afap1L1a to determine the subcellular localization of the protein. (**B**) Ubiquitous expression of the *afap1l1a* RNA in 1, 6 and 10-hpf embryos. (**C**) A single plane of PGCs expressing EGFP-F′ and the mCherry-Afap1L1a fusion protein reveals weak expression of the protein around the cell perimeter (**C′**) and a predominant strong association with the base of filopodia (**C′–C′′′**). In **C′**, one marks the site of a retracted filopodium, two the sites of extended filopodia and arrow indicates the direction of migration. (**D**) The number and distribution of filopodia in migrating PGCs overexpressing *afap1l1a* (dark bars) as compared with control PGCs (light bars). 'n' indicates the number of cells analysed. (**E**) Representative images of control (left) and of *afap1l1a*-overexpressing PGCs. Arrows indicate the direction of movement and arrowheads point at filopodia. Scale bars signify 10 μm except for **C′′** and **C′′′**, where the scale is 3 μm.

The following figure supplements are available for figure 7:

**Figure supplement 1**. The effect of Afap1L1a overexpression on filopodia dynamics in PGCs migrating within a wild type environment.

**Figure supplement 2**. Afap1L1a overexpression does not affect bleb formation at the cell front.

important step in normal and pathological conditions where migrating cells initiate their reaction to guidance signals.

## Materials and methods

### Zebrafish strains

Zebrafish (*Danio rerio*) of the AB background and transgenic fish carrying the *kop-egfp-f-nos-3′UTR* transgene (*Blaser et al., 2005*), *kop-mcherry-f-nos3′UTR* transgene, or the *kop-lifeact-egfp-nos3′UTR* transgene were used as wild type fish. *medusa^{NYO45}* (*Valentin et al., 2007*) mutant embryos were used for investigating filopodia formation in the absence of the guidance cue Cxcl12a and odysseus (ody) homozygous mutant fish were used for the rescue experiment with Cxcr4b.

### RNA expression constructs and injections

A list of constructs and in situ probes of the Zebrafish genes generated in this work, containing primers and amounts injected, is provided in *Supplementary file 1B*. Commonly PGCs were labelled with farnesylated mCherry (mCherry-F′) or EGFP (EGFP-F′). The dominant-negative form of Irsp53 was generated according to Millard et al. (*Millard et al., 2005*) Additional constructs with amounts injected are found in *Supplementary file 1C*.

To direct protein expression to the germ cells, the corresponding open reading frames (ORFs) were cloned upstream to the 3′UTR of the *nanos*1 gene, facilitating translation and stabilization of the RNA in these cells (*Koprunner et al., 2001*). For global protein expression in the embryo, the ORFs were cloned upstream of the 3′UTR of the *Xenopus globin* gene.

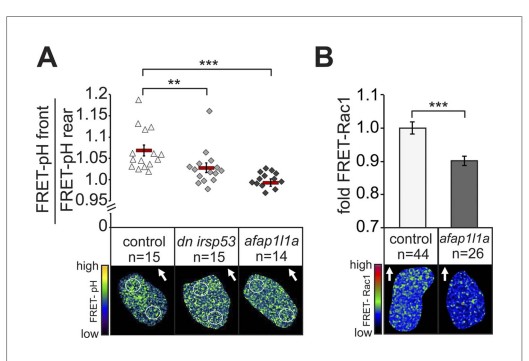

**Figure 8**. Filopodia are required for cellular response to polarized Cxcl12a distribution. (**A**) 6–7 hpf control PGCs exhibit polarized distribution of intracellular pH as determined by the FRET efficiency of the pH sensor pH-lameleon5 protein in the cells (left). Expression of the dominant negative form of Irsp53 or overexpression of the Afap1L1a abrogates the formation of high pH in the front. The graph represents average pH-FRET ratios between the front and the rear (as indicated by the circles). The values for each cell are averages of 20 time points. (**B**) 8–9 hpf PGCs overexpressing Afap1L1a exhibit a decrease in Rac1 activity, as determined by differences in FRET generated by a Rac1-FRET activity reporter. Arrows indicate the direction of movement. 'n' indicates the number of cells analysed.

The following figure supplement is available for figure 8:

**Figure supplement 1**. Filopodia formation is independent of the elevated pH at the cell front and of Rac1 activity in migrating PGCs.

Capped sense mRNA was synthesized using the mMessage Machine kit (Ambion, Foster City, CA). 2 nanoliters of RNA and/or morpholino antisense oligonucleotides were microinjected into the yolk of 1-cell stage embryos, unless stated otherwise.

### Morpholino knockdown experiments

To inhibit protein translation in the embryo, morpholino antisense oligonucleotides (Gene Tools, Philomath, OR) were injected into the one-cell stage embryos. A list of morpholinos used in this work is provided in *Supplementary file 1D*.

### In situ hybridization

Whole-mount in situ hybridization was performed as previously described (*Weidinger et al., 1999*), using *afap1l1a* and *irsp53* digoxigenin-labeled probes.

The sequence of the primers used for PCR amplification of these probes from zebrafish cDNA is provided in the *Supplementary file 1B*.

### Transplantation experiments

For the design of the experiment and amount of the injected material see *Supplementary file 1A–C*. Cells from a 4 hpf donor embryo were transplanted into a 5–6 hpf host on a Zeiss AxioImager.M2 microscope with a 5× objective, immediately followed by the acquisition of a time-lapse video using a 40× objective or followed by acquisition of snapshots using a 63× objective.

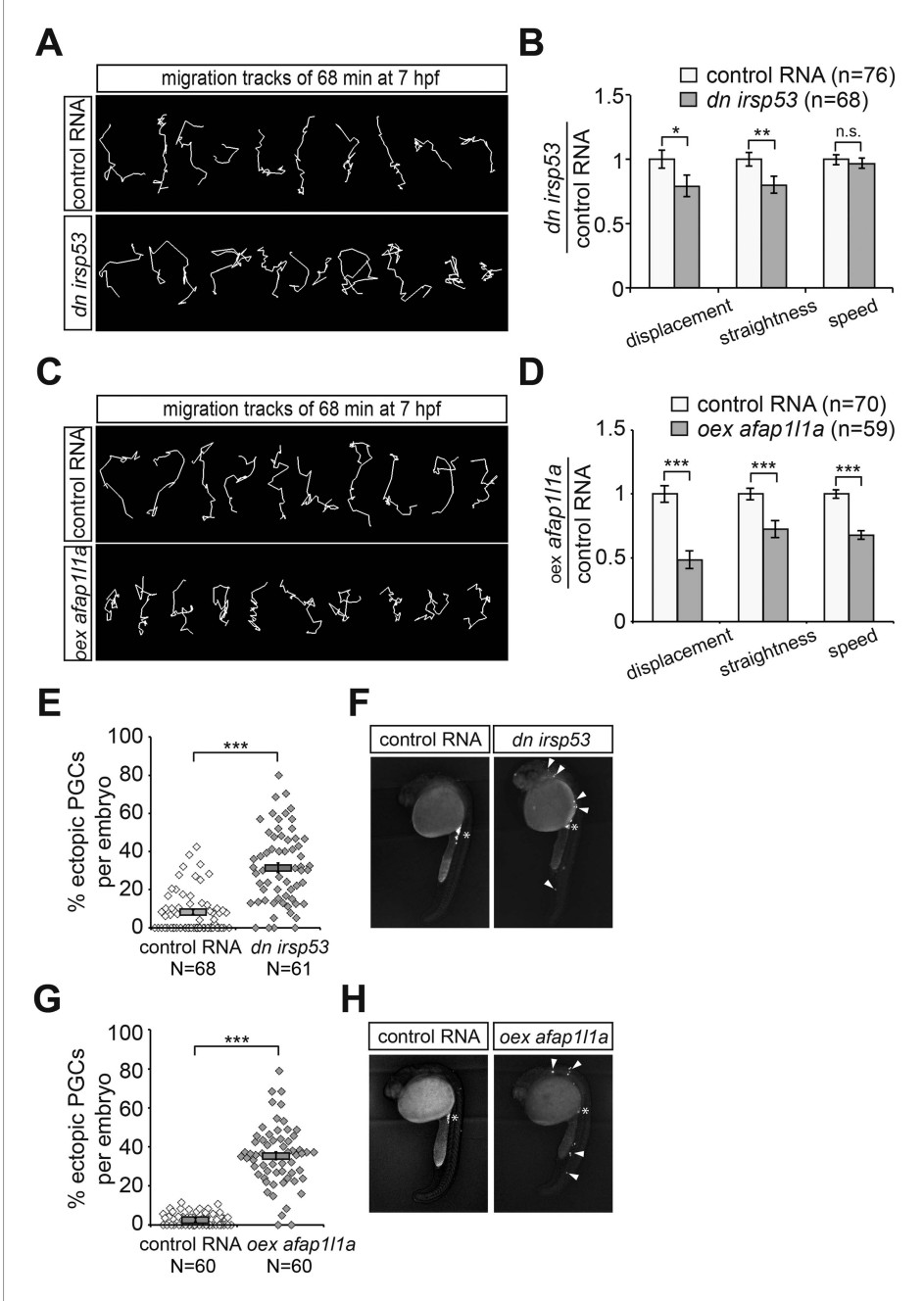

**Figure 9**. Manipulations of filopodia formation lead to PGC migration defects. (**A**) Representative migration tracks of control PGCs (upper panel) and PGCs expressing the dominant-negative (*dn*) *irsp53* version (lower panel). (**B**) Analysis of PGC migration tracks assessing displacement, straightness and migration speed, comparing control cells (light bars) with *dn Irsp53*-expressing cells (dark bars). 'n' indicates the number of migration tracks analysed. (**C**) Representative migration tracks of control PGCs (upper panel) and PGCs overexpressing (*oex*) *afap1L1a* (lower panel). (**D**) Analysis of PGC displacement, track straightness and migration speed, comparing control cells (light bars) with *afap1l1a* overexpressing cells (dark bars). 'n' indicates the number of migration tracks analysed. (**E–H**) Expression of the *dn irsp53* (**E**, **F**) or overexpressing *afap1l1a* (**G**, **H**) in PGCs results in an increase of ectopic cells at 24 hpf as compared with embryos whose PGCs express a control RNA. 'n' indicates the number of embryos analysed. Arrowheads point at ectopic PGCs and asterisks mark the site of the developing gonad.

The following figure supplement is available for figure 9:

**Figure supplement 1**. Reducing the activity of Irsp53 has no effect on cell migration in the absence of Cxcl12a.

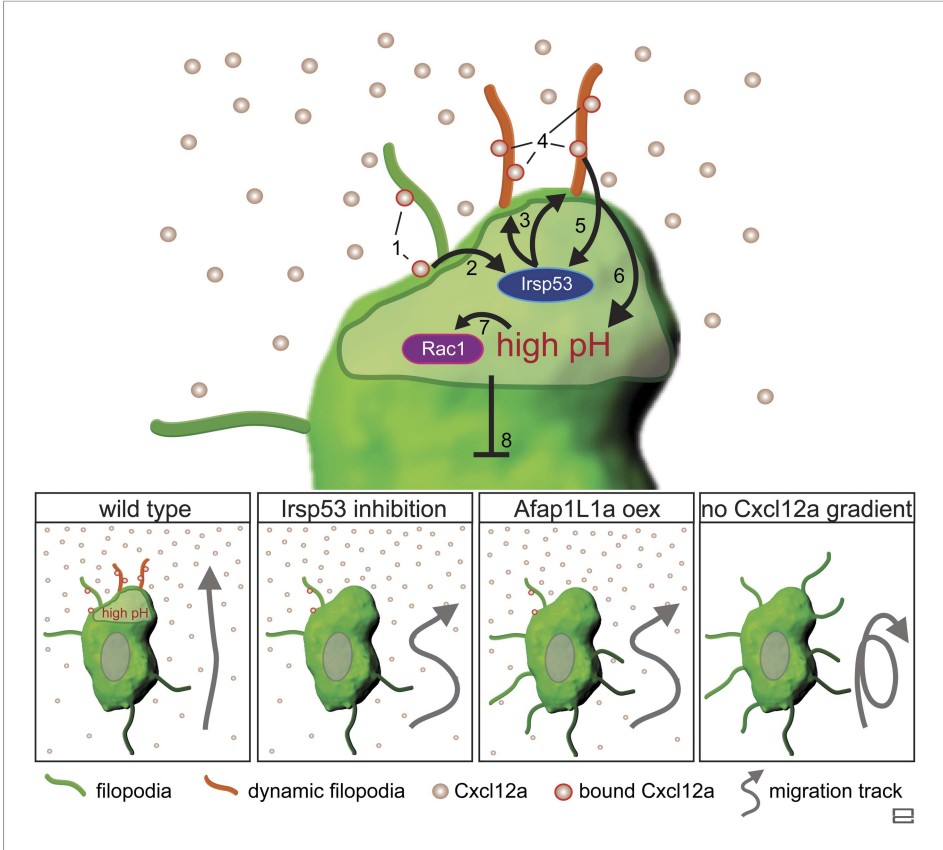

**Figure 10**. Regulation of dynamic filopodia at the cell front and their role in cell polarization and directed cell migration. Following the activation of Cxcr4b by Cxcl12a (1), the scaffold protein Irsp53 is activated (2) and promotes the formation of dynamic filopodia at the cell front (3). The dynamic filopodia that extend in the direction of higher Cxcl12a concentration increase the surface for chemokine binding (4), thereby enhancing signalling. The enhanced signalling at the cell front results in further Irsp53 activation (5) and an elevation of pH (6). The increase in pH in turn, leads to an elevation in Rac1 activity at this aspect of the cell (7). Once established, the front inhibits other parts of the cell from assuming front characteristics including dynamic filopodia formation, elevated pH and Rac1 activation (8). Lower panels—Irsp53 inhibition or Afap1L1a overexpression result in loss of dynamic filopodia at the cell front, abrogating the local increase in pH. As a consequence, PGCs migrate less directionally. Lack of a Cxcl12a gradient leads to the formation of numerous, evenly distributed, long and persistent filopodia in cells that migrate in random directions.

## Image acquisition

Time-lapse imaging was performed using a Zeiss AxioImager.M2 microscope equipped with a dual view filter (MAG Biosystems, Exton, PA), Photometrics cameras (Cascade II and CoolSNAP ES2) and VS-Laser Control.

Time-lapse videos were generated using 63×- or 40× water-immersion objectives for imaging cell morphology and behaviour, and a 10× objective for the purpose of speed and migration track analysis. Detailed experimental setups are listed in *Supplementary file 1A*. Experiments were performed between 6–9 hpf.

Frames were captured simultaneously with 488 nm and 561 nm lasers at 10-or 12 s intervals for high magnification videos (63×) and at 15-s intervals for 40× videos, with 300 ms exposure time. 24 to 37 images at focal planes 1 µm (for 63× videos) or 4 µm (for 40× videos) apart were captured to generate the Z-stacks. For migration track and speed analysis time-lapse videos were captured with a 10× objective at 2-min intervals over 68 min with 300 ms exposure time at three focal planes (15 µm apart) to generate the Z-stacks.

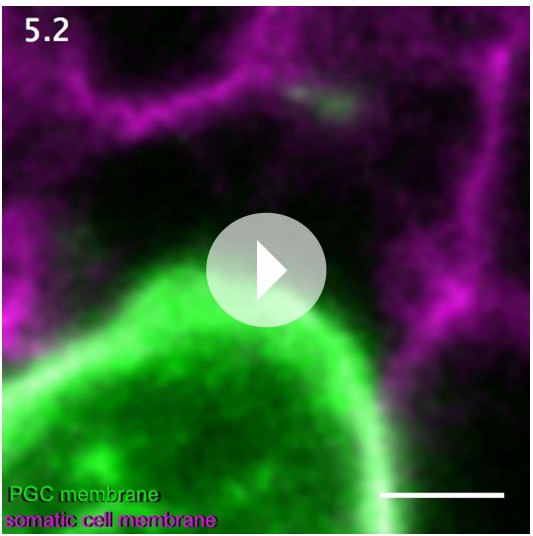

**Video 10.** Some filopodia adhere to somatic cells and appear to be under tension. Laser-induced cut of a filopodium was performed using a Zeiss LSM710 confocal microscope equipped with a Zeiss W Plan-Apochromat 63× objective controlled by the ZEN software (Zeiss, Germany). Following cut, the tip of the filopodium remained in contact with the distant somatic cell, while the part of the protrusion attached to the cell rapidly retracted, signifying tension. The video spans 7 s with snapshots captured at a 200 ms time-interval (pinhole 208 μm, 128 × 128). Time indicated in seconds and scale bar represents 5 μm.

## Data analysis

For filopodia quantification the first time-point of a 2-min video was analysed and filopodia dynamics was followed over a time-period of 10 min. For filopodia measurements the manual mode of the 'filament tracer' module of the Imaris software (Bitplane, Switzerland) was used. PGC response to transplants was captured for 30 min starting ca. 2 min after transplantation. To calculate the percentage of ectopic germ cells, PGCs were counted at 24 hpf in the GFP-channel. The average number of ectopic PGCs was counted in percentage to the average total PGC number for the given experiment.

For gastrulation correction, speed measurements and tracking of migrating germ cells the 'spots' module of the Imaris software (Bitplane, Switzerland) was used. Analysis of Cxcr4b distribution and protein age on the PGC membrane was done using ImageJ software (for the analysis protocol see *Supplementary file 1F*).

## FRET analysis

Imaging and analysis for the Rac1-FRET (as snapshots) was performed as for the pH-FRET (as time-lapse). To measure the pH distribution in PGCs, *pH-lameleon5* (*Esposito et al., 2008*) mRNA fused to the 3′UTR of *nanos1* was injected into 1-cell stage of *kop-mCherry-f-nanos3′UTR* transgenic embryos. Imaging was performed at 7–9 hpf using an LSM710 confocal microscope (40×, NA 0.75, pinhole 276 μm, 512 × 512, 7.5 s per frame) controlled by the ZEN software (Zeiss, Germany). Analysis was done using ImageJ software (for the analysis protocol see Supporting material).

## Cxcl12a visualization

Embryos that express mCherry on the membrane of PGCs were knocked down for Cxcr7b using a morpholino antisense oligonucleotide by injection at 1-cell stage and were then injected with *cxcl12a-venus* RNA into a corner cell of the 16-cell stage for a mosaic expression pattern of the chemokine. Time-lapse imaging was performed at 10–15 hpf using either a Zeiss AxioImager.M2 microscope (63× water imersion objective), equipped with a dual view filter (MAG Biosystems, Exton, PA), Photometrics cameras (Cascade II and CoolSNAP ES2) and VS-Laser Control (*Video 8*) or a Lightsheet Z1 (Zeiss, Germany) equipped with a Zeiss W Plan-Apochromat 20× objective (15 s intervals for 20 cycles at 960 × 960) and controlled by the ZEN software (Zeiss, Germany) (*Video 9*).

## Laser ablation experiments

Laser ablation was performed with a Zeiss LSM710 confocal microscope equipped with a Zeiss W Plan-Apochromat 63× objective controlled by the ZEN software (Zeiss, Germany). The 'bleaching' module of the Zen software was used to perform ablation within a sub-region (ROI). The laser source for ablation was a femtosecond Titanium-Sapphire laser (Coherent Chameleon, Santa Clara, CA) tuned to lambda = 740 nm with laser power reduced to 10%. The power at the objective back aperture during ablation was 90–100 mW. Ablation was monitored with confocal imaging of PGC membrane (laser 488 nm) and somatic membrane (laser 561 nm) in *med*[NY054] homozygous embryos over 45 s with 199 ms time-interval (pinhole 208 μm, 128 × 128).

## Statistical analysis

All the data in the work were first tested for normal distribution by the Kolmogorov–Smirnov test. Experiments on the percentage of ectopic germ cells and FRET measurements were analysed by the Student's t-test. Experiments on the signal intensity between front and back of a cell were analysed using the paired t-test. All other significance tests are based on the Mann Whitney U test for the statistical difference. Error bars represent S.E.M. n.s. = non significant, $*p \leq 0.05$, $**p \leq 0.01$, $***p \leq 0.001$.

## Acknowledgements

We acknowledge the support of the University of Münster, the Cells in Motion (CIM) excellence cluster, the German research foundation (DFG) and the European Research Council (ERC). We thank Michael Knop for the tandem timer construct, Ina Halbig, Ursula Jordan and Ines Sandbote for technical support and Nina Gerigk for graphical help.

## Additional information

### Funding

| Funder | Grant reference | Author |
| --- | --- | --- |
| Deutsche Forschungsgemeinschaft (DFG) | | Dana Meyen, Carolina Wittwer |
| Westfälische Wilhelms-Universität Münster | Cells in Motion (CIM) excellence cluster | Torsten U Banisch |
| European Research Council (ERC) | | Katsiaryna Tarbashevich, Benoît Maugis |

The funders had no role in study design, data collection and interpretation, or the decision to submit the work for publication.

### Author contributions

DM, Conception and design, Acquisition of data, Analysis and interpretation of data, Drafting or revising the article, Contributed unpublished essential data or reagents; KT, TUB, CW, Acquisition of data, Analysis and interpretation of data; MR-F, Drafting or revising the article, Contributed unpublished essential data or reagents; BM, Performed filopodia ablation for Video 10; CG, Carried out experiments aimed at determining the role of Cxcl12 in filopodia formation; E-MM, Scored Afap1l1a ectopic PGCs in Figure 9G With DM, Acquisition of data; ER, Conception and design, Analysis and interpretation of data, Drafting or revising the article

### Author ORCIDs

Torsten U Banisch, http://orcid.org/0000-0002-9995-1580

### Ethics

Animal experimentation: This study was performed in accordance with the regulation in the State of North-Rhine Westphalia, Germany, under the supervision of the veterinarian director Dr. Otto in Münster (Amt für Gesundheit, Veterinär- und Lebensmittelangelegenheiten, 39.32.7.1).

## Additional files

### Supplementary file

• Supplementary file 1. Experimental design, additional materials and protocols.

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
