## [Decision Letter]

Thank you for sending your work entitled “Dynamic filopodia are required for chemokine-dependent intracellular polarization during guided cell migration in vivo” for consideration at *eLife*. Your article has been favorably evaluated by K VijayRaghavan (Senior editor), Marianne Bronner (Reviewing editor), and 3 reviewers.

The Reviewing editor and the reviewers discussed their comments before we reached this decision, and the Reviewing editor has assembled the following comments to help you prepare a revised submission.

Meyen et al. examine migration of PGCs in zebrafish to unravel the question of how migration induced by Cxcl signaling is controlled by filopodia. The main conclusion of the manuscript is that filopodia distribution and their dynamics are dictated by a gradient of the chemokine Cxcl12a. Anterior filopodia, in turn are required for receiving the cxcl12a signal that then polarizes the cell. They also conclude that there are two kinds of filopodia: some that are chemokine dependent and others that are not. This manuscript focuses on an interesting topic and up to Figure 4, the experiments are clear and the interpretation of the results is sounds. The work is potentially appropriate for *eLife* after the following issues have been addressed:

1) The authors find a correlation between how dynamic a filopodia is and the presence of the chemokine CXCL12. The authors suggest that CXCL12 influence filopodia dynamic; however CXCL12 controls directional migration as well; therefore it is possible that directional migration could affect filopodia dynamics. In other words if a cell is moving in a particular directions the filopodia formed in that direction (front) will be short lived because they will be engulfed by the cell or cell protrusions (blebs in this case, example in Video 8). Are the front filopodia more dynamic because they disappear as they are engulfed by blebs during migration? Or do they maintain the length at the front in spite the cells is moving in the direction of the filopodia? This issue can be easily addressed analyzing filopodia dynamics in non-motile cells exposed to CXCL12.

2) The authors claim: “These findings are consistent with the idea that filopodia formation constitutes an early response to distribution of Cxcl12 that is followed by cell polarization and directed migration”. To support this claim the authors need to analyze the sequence of events in wild type PGC: are the filopodia really appearing earlier than polarization and directed migration? (This is not obvious in Video 6).

3) The authors show no correlation between the presence of filopodia and the formation of blebs in control cells (Figure 2—figure supplement 3). A similar analysis should be performed for the cells in which filopodia has been affected (DN Irsp53 and Afap1L1a-overexpressing cells). This analysis is important to support their conclusion that filopodia affect CXCL12 sensing, as they need to rule out the alternative explanation that filopodia interfering is affecting directional migration by disturbing bleb formation.

4) Increased Cxcl12a leads to increased pH at the cell front. Expression of DN-Irsp53 or AFAP1L1 blocked the increased pH. Why did the AFAP1L1 not lead to increased pH at the back? Given their model, shouldn't this predict increased filopodia at the back?

5) The authors suggest filopodia function in controlling intracellular pH at the front, and are formed independent of pH. Based on their model, increased filopodia leading to increasing sensitivity to Cxcl12a, would predict that increased filopodia in AFAP1L overexpression might make cells more sensitive to a Cxcl12 gradient. Can this be quantitated?

6) One of the main conclusions of the paper is the existence of different types of filopodia. This should not only be discussed in the Discussion section but already in the Abstract or Introduction and Results sections.

Figure 5: Is Irsp53 expressed in all filopodia? Expression of DN Irsp53 only affects front and side filopodia (5D). Surprisingly, the overexpression of DN Irsp53 in *medea* mutant embryos has no effect on filopodia arguing that *medea* only has filopodia of the posterior/chemokine-independent type. This should be mentioned in the Results section and also further tested (see below).

Do the authors believe that the filopodia that form in *medea* after cxcl12a overexpression now possess an anterior character? The subsection headed “Filopodia formation and dynamics are regulated by the chemokine Cxcl12” seems to suggest that they still look like the typical posterior type *medea* filopodia. However, if the authors' claim is correct that anterior filopodia form because of exposure to Cxl12a and subsequent activation of lrsp53, filopodia in medea embryos that overexpress Cxl12a should now all look like anterior filopodia. This should be tested. Likewise, the overexpressing dnLrsp53 should be able to deplete filopodia in Cxcl12a mRNA injected *medea* mutant embryos. Also, how does manipulation of Afap1L1 affect filopodia in *medea* mutant embryos?

7) The authors describe that 50% of posterior filopodia do not contain actin (in the subsection headed “Migrating PGCs extend specialized filopodia in the direction of migration”). They do not describe how they interpret this finding. Does this argue for the presence of two different types of filopodia? But then there are still 50% of filopodia with a front phenotype expressed in the back. What does that imply? Are the actin-less filopodia responsive to Cxcl12a? Do filopodia in *medea* mutants all possess actin? Does the absence or presence of actin have anything to do with chemokine signaling?

8) The authors propose that filopodia increase the Cxcl12a signal by increasing the surface area for ligand binding at the front. This is an interesting idea, but difficult to prove. Is Cxcl12 binding preferentially to the filopodia at the front and not to those formed at the back? This could be easily analysed by repeating the analysis performed in Figure 4, but now in filopodia formed at the back, or in those conditions where the type of filopodia (back or front) has been altered.

---

## [Author Response]

*1) The authors find a correlation between how dynamic a filopodia is and the presence of the chemokine CXCL12. The authors suggest that CXCL12 influence filopodia dynamic; however CXCL12 controls directional migration as well; therefore it is possible that directional migration could affect filopodia dynamics. In other words if a cell is moving in a particular directions the filopodia formed in that direction (front) will be short lived because they will be engulfed by the cell or cell protrusions (blebs in this case, example in*
Video 8*). Are the front filopodia more dynamic because they disappear as they are engulfed by blebs during migration? Or do they maintain the length at the front in spite the cells is moving in the direction of the filopodia? This issue can be easily addressed analyzing filopodia dynamics in non-motile cells exposed to CXCL12*.

We thank the referees for raising this point, which we addressed by examining the fate of filopodia in non-manipulated cells. We wished to avoid possible side effects caused by rendering the cells immotile since for example, inhibiting motility of germ cells may interfere with cellular responses that in turn may impinge on filopodia dynamics. To this end, we examined the persistence of individual filopodia in 10-minute time-lapse movies, while monitoring for engulfment of filopodia by advancing blebs. First, only a third of the filopodia that extended at the cell front, are engulfed by inflating blebs. Still, the persistence of the remaining front filopodia in cells guided by physiological Cxcl12 gradients is lower than that of filopodia at the back of the cell. Those results are now presented in Figure 3—figure supplement 1 and are described in the text. Thus, whereas blebs can indeed engulf filopodia thereby affecting their persistence, the rest of the filopodia at the cell front exhibit shorter life span relative to those at the cell rear.

*2) The authors claim:* “*These findings are consistent with the idea that filopodia formation constitutes an early response to distribution of Cxcl12 that is followed by cell polarization and directed migration*”*. To support this claim the authors need to analyze the sequence of events in wild type PGC: are the filopodia really appearing earlier than polarization and directed migration? (This is not obvious in*
Video 6*)*.

We indeed consider the sequence of events gradient->filopodia->change in cell polarization a central claim in the paper. To make this point crystal clear, we made use of the golgi as a directional marker since in migrating germ cells, the golgi resides behind the nucleus and thus marks the back of the cell. We now present the region of the golgi by using farnesylated –EGFP (red asterisk in Figure 4—figure supplement 1) in the movie (red asterisk in current Video 6, Cxcl12a transplant) and in the presented snapshots based on individual confocal planes (Figure 4). This sequence of events is observed in the vast majority of cells responding to a “simple” artificial gradient introduced into the embryonic environment lacking Cxcl12 (15 times of 17 cases examined) and this point is mentioned in the text.

Cells in non-manipulated embryos experience a more complex set of instructions, as the endogenous gradients are shallow and not as focused. Nevertheless, as requested we now present the sequence of events as recorded in cells performing sharp turns in the context of a non-manipulated embryo (Video 7 and snapshots presented in Figure 4—figure supplement 2). Those new results are now described in the text.

*3) The authors show no correlation between the presence of filopodia and the formation of blebs in control cells (*Figure 2—figure supplement 3*). A similar analysis should be performed for the cells in which filopodia has been affected (DN Irsp53 and Afap1L1a-overexpressing cells). This analysis is important to support their conclusion that filopodia affect CXCL12 sensing, as they need to rule out the alternative explanation that filopodia interfering is affecting directional migration by disturbing bleb formation*.

In the data we provide in Figure 2—figure supplement 3, we show that the formation of the blebs and filopodia is not coupled when employing tighter temporal and spatial criteria, suggesting that filopodia are not required at the actual time and place of bleb formation. To address the referee comment, we further show now that the manipulations aimed at specifically altering the distribution of the filopodia do not affect other relevant cellular features such as bleb formation. We monitored this parameter in DN Irsp53 and Afap1L1a-overexpressing cells. Our findings now are presented in Figure 6—figure supplement 2 (DN Irsp53) and Figure 7—figure supplement 2 (Afap1L1a overexpression) and show no significant effect on the frequency of bleb formation in the cells in which filopodia formation is impaired.

*4) Increased Cxcl12a leads to increased pH at the cell front. Expression of DN-Irsp53 or AFAP1L1 blocked the increased pH*. *Why did the AFAP1L1 not lead to increased pH at the back? Given their model, shouldn't this predict increased filopodia at the back?*

We would like to explain why our model does not predict elevation of pH or increased dynamic filopodia formation at the back of germ cell upon increase of filopodia at the site. Wild type PGCs that sense a gradient of Cxcl12a respond by elevation of the pH at the cell front, which leads to an increase in Rac1 activity at this location. Manipulating the distribution of the filopodia, such that their number at the cell front is reduced, adversely affects the efficiency with which the PGCs can perceive the chemokine signal and interpret it as a graded one. As a consequence, the establishment of higher pH at the cell front is hindered. Afap1L1 overexpression leads to a relative increase in filopodia number at the back of cells. Those cells encounter higher levels of the chemokine at their front, but are defective in perceiving the signal. The back aspect of the cell, while possessing more filopodia encounters less of the chemokine. We thus interpret the results such that Afap1L1 overexpressing cells with “less efficient front-more efficient back” simply fails to form the Cxcl12a-dependent intracellular polarization in the distribution of the pH. The increase in filopodia in the back did not win over the front but rather cancelled out the cell polarity that is normally induced by the chemotactic signal in the front.

5) The authors suggest filopodia function in controlling intracellular pH at the front, and are formed independent of pH. Based on their model, increased filopodia leading to increasing sensitivity to Cxcl12a, would predict that increased filopodia in AFAP1L overexpression might make cells more sensitive to a Cxcl12 gradient. Can this be quantitated?

As described in our response to the previous point, cells overexpressing AfapL1a do not simply possess more filopodia, but actually loose the normal polarity where more dynamic filopodia form at the cell front. The loss of the normal arrangement/specific features of the filopodia interferes with the ability of the cell to interpret the Cxcl12a gradient, which is the basis for the miss-migration phenotype we observe under these conditions.

*6) One of the main conclusions of the paper is the existence of different types of filopodia. This should not only be discussed in the Discussion section but already in the Abstract or Introduction and Results sections*.

In the current version of the paper this point i.e., the fact that actin-rich filopodia that show higher dynamics and higher actin content are formed at the front of cells migrating within a Cxcl12a gradient, is now mentioned at the end of the Introduction.

Figure 5*: Is Irsp53 expressed in all filopodia? Expression of DN Irsp53 only affects front and side filopodia (5D). Surprisingly, the overexpression of DN Irsp53 in* medea *mutant embryos has no effect on filopodia arguing that* medea *only has filopodia of the posterior/chemokine-independent type. This should be mentioned in the Results section and also further tested (see below)*.

Based on the detection of an Irsp53-mCherry fusion protein, about 40% of the filopodia at the cell front contain Irsp53, while a lower proportion of filopodia in other regions of the cell contain the protein. We present these new quantitative data in Figure 6—figure supplement 1. In contrast, the dominant-negative version of the protein is found primarily in the cell body rather than in the filopodia (Figure 6—figure supplement 1). This issue is now mentioned in the Results section.

Indeed, the filopodia formed in the Cxcl12 mutant *medusa* are chemokine independent (by definition) and appear “back-like” and show “back-like” characteristics including dynamics and length properties, similar to those of back filopodia in wild type cells. Consistently, expressing the dominant-negative Irsp53 in PGCs of *medusa* embryos does not affect the filopodia, since Irsp53 is expressed primarily within and is required for front filopodia formation. As requested, we now mention this point in the Results section.

*Do the authors believe that the filopodia that form in* medea *after cxcl12a overexpression now possess an anterior character? The subsection headed “Filopodia formation and dynamics are regulated by the chemokine Cxcl12” seems to suggest that they still look like the typical posterior type* medea *filopodia. However, if the authors' claim is correct that anterior filopodia form because of exposure to Cxl12a and subsequent activation of lrsp53, filopodia in medea embryos that overexpress Cxl12a should now all look like anterior filopodia. This should be tested. Likewise, the overexpressing dnLrsp53 should be able to deplete filopodia in Cxcl12a mRNA injected* medea *mutant embryos. Also, how does manipulation of Afap1L1 affect filopodia in* medea *mutant embryos?*

We find polarization in the filopodia types formed by cells only when the cells are presented with graded Cxcl12a in the environment. Simply exposing cells to uniform Cxcl12a, or having no Cxcl12a in the environment leads to the formation of filopodia that are less dynamic, despite the fact that cells do migrated albeit non-directionally under these conditions. This conclusion is based on our experiments, where we follow filopodia formation in *medusa* embryos, as well as in embryos in which Cxcl12a was introduced globally by RNA injection into early embryos (Figure 3). In this experimental set up the forming filopodia exhibited uniform “back” properties regardless of the region of the cells where they arose. Since in the absence of a Cxcl12a gradient anterior-type filopodia do not form, DN Irsp53 is not expected to affect them under these conditions. Indeed, expression of DN Irsp53 in germ cells in *medusa* embryos overexpressing Cxcl12a did not result in reduction of filopodia number, the effect that is normally observed in wild type cells. See Figure 11.

Author response image 1.**DOI:**
http://dx.doi.org/10.7554/eLife.05279.042

*Also, how does manipulation of Afap1L1 affect filopodia in* medea *mutant embryos?*

The filopodia in PGCs in *medusa* embryos exhibit dynamic and structural properties characteristic of “back filopodia”. In wild type embryos, Afap1L1 overexpression results in a reduction in the front filopodia and in an increase in the number of “back-type” filopodia.

From experiments conducted in response to this question (see Figure 12), overexpression of Afap1L1 had no effect on filopodia properties beyond those already conferred by the absence of the chemokine Cxcl12a (=the conditions in *medusa* embryos).

Author response image 2.**DOI:**
http://dx.doi.org/10.7554/eLife.05279.043

*7) The authors describe that 50% of posterior filopodia do not contain actin (in the subsection headed “Migrating PGCs extend specialized filopodia in the direction of migration”). They do not describe how they interpret this finding. Does this argue for the presence of two different types of filopodia? But then there are still 50% of filopodia with a front phenotype expressed in the back. What does that imply? Are the actin-less filopodia responsive to Cxcl12a? Do filopodia in* medea *mutants all possess actin? Does the absence or presence of actin have anything to do with chemokine signaling?*

In a significant fraction of the posterior filopodia we failed to detect lifeact signal. We believe that some back filopodia contain lower concentration of actin rather than completely devoid of it. We have reviewed all sites in the text where we mention these structures and made sure the point is understood in this way. We interpret the results to indicate that front filopodia contain a higher concentration of actin, such that it is always detected but have no information concerning an actual functional significance of the difference. We consider however low-actin filopodia a characteristic of back filopodia. As expected from the more posterior nature of filopodia formed in *medusa* mutants, the fraction of filopodia in which no lifeact could be detected is indeed elevated (now presented in Figure 2—figure supplement 1).

*8) The authors propose that filopodia increase the Cxcl12a signal by increasing the surface area for ligand binding at the front. This is an interesting idea, but difficult to prove. Is Cxcl12 binding preferentially to the filopodia at the front and not to those formed at the back? This could be easily analysed by repeating the analysis performed in*
Figure 4*, but now in filopodia formed at the back, or in those conditions where the type of filopodia (back or front) has been altered*.

We indeed believe that the increased number of filopodia at the cell front plays a role in polarizing the cell in response to the ligand, by increasing the chance Cxcl12a interacts with the cell at the front. In the experiments aimed at demonstrating the function of filopodia we detect small aggregates of Venus-labeled ligand interacting with filopodia and getting internalized (Figure 5). We do not believe the back filopodia cannot function in Cxcl12a-binding, since the receptor Cxcr4b is uniformly expressed around the cell circumference. Indeed, in data we provide in the revised version of the work (Figure 5—figure supplement 1) ligand binding to back filopodia can be observed as well. This finding is now mentioned in the Results section.